# Sub-THz wireless transmission based on graphene-integrated optoelectronic mixer

Alberto Montanaro [1,2] ✉, Giulia Piccinini[3,4], Vaidotas Mišeikis[4,5], Vito Sorianello [1], Marco A. Giambra[6], Stefano Soresi[6], Luca Giorgi[7], Antonio D'Errico [7], K. Watanabe [8], T. Taniguchi [9], Sergio Pezzini [10], Camilla Coletti [4,5] & Marco Romagnoli [1]

Optoelectronics is a valuable solution to scale up wireless links frequency to sub-THz in the next generation antenna systems and networks. Here, we propose a low-power consumption, small footprint building block for 6 G and 5 G new radio wireless transmission allowing broadband capacity (e.g., 10–100 Gb/s per link and beyond). We demonstrate a wireless datalink based on graphene, reaching setup limited sub-THz carrier frequency and multi-Gbit/s data rate. Our device consists of a graphene-based integrated optoelectronic mixer capable of mixing an optically generated reference oscillator approaching 100 GHz, with a baseband electrical signal. We report >96 GHz optoelectronic bandwidth and −44 dB upconversion efficiency with a footprint significantly smaller than those of state-of-the-art photonic transmitters (i.e., <0.1 mm$^2$). These results are enabled by an integrated-photonic technology based on wafer-scale high-mobility graphene and pave the way towards the development of optoelectronics-based arrayed-antennas for millimeter-wave technology.

The fast rise of data-hungry and performance-demanding applications, e.g., digital twins, AI computing, autonomous driving, robotic surgery, digital biology, call for the need for novel access technology. Advanced 5 G and 6 G will make use of New Radio (NR) access technology[1]. While 5 G keeps its promises, advanced 5 G has more potential to implement[2]. 5 G NR access operates in the sub-1 GHz to 100 GHz spectrum[2], and supports various use cases, including time-critical services for consumers, enterprises, and public institutions across multiple sectors[1,2]. NR is currently a standard in the 3rd Generation Partnership Project (3GPP)[2], and a new generation of antenna systems is being developed[3]. In the last decade, mobile networks have increased carried traffic by almost 300 times, and network speeds have

increased hundreds of times[1]. In the journey toward 6 G, new extremely demanding and immersive applications will require a further upgrade of signal processing functions with improved signal integrity, frequency, and capacity[2]. The need for broader bandwidth capacities in wireless networks can be satisfied by increasing the carrier frequency (CF). Therefore, the sub-THz frequency range (90–300 GHz) is becoming more and more appealing[4–6] and, given the potential to provide hundreds of Gbit/s over short distances, it will be a key enabler for 6 G systems[4]. Figure 1 depicts a scenario including three types of antenna systems, envisioned for sub-THz transmissions. The first is a single antenna transmitter, used in high throughput point-to-point radio links with application in the backhaul transport network[7] (Fig. 1,

[1]Photonic Networks and Technologies Lab – CNIT, Via G. Moruzzi,1, 56124 Pisa, Italy. [2]TeCIP Institute, Scuola Superiore Sant'Anna, via G. Moruzzi 1, 56124 Pisa, Italy. [3]NEST, Scuola Normale Superiore, Piazza San Silvestro 12, 56127 Pisa, Italy. [4]Center for Nanotechnology Innovation @NEST, Istituto Italiano di Tecnologia, Piazza San Silvestro 12, 56127 Pisa, Italy. [5]Graphene Labs, Istituto Italiano di Tecnologia, Via Morego 30, 16163 Genova, Italy. [6]Inphotec, CamGraPhIC srl, via G. Moruzzi 1, 56124 Pisa, Italy. [7]Ericsson Research, via G. Moruzzi 1, 56124 Pisa, Italy. [8]Research Center for Electronic and Optical Materials, National Institute for Materials Science, 1-1 Namiki, Tsukuba 305-0044, Japan. [9]Research Center for Materials Nanoarchitectonics, National Institute for Materials Science, 1-1 Namiki, Tsukuba 305-0044, Japan. [10]NEST, Istituto Nanoscienze-CNR and Scuola Normale Superiore, P.zza S. Silvestro 12, 56127 Pisa, Italy. ✉e-mail: amontanaro@cnit.it

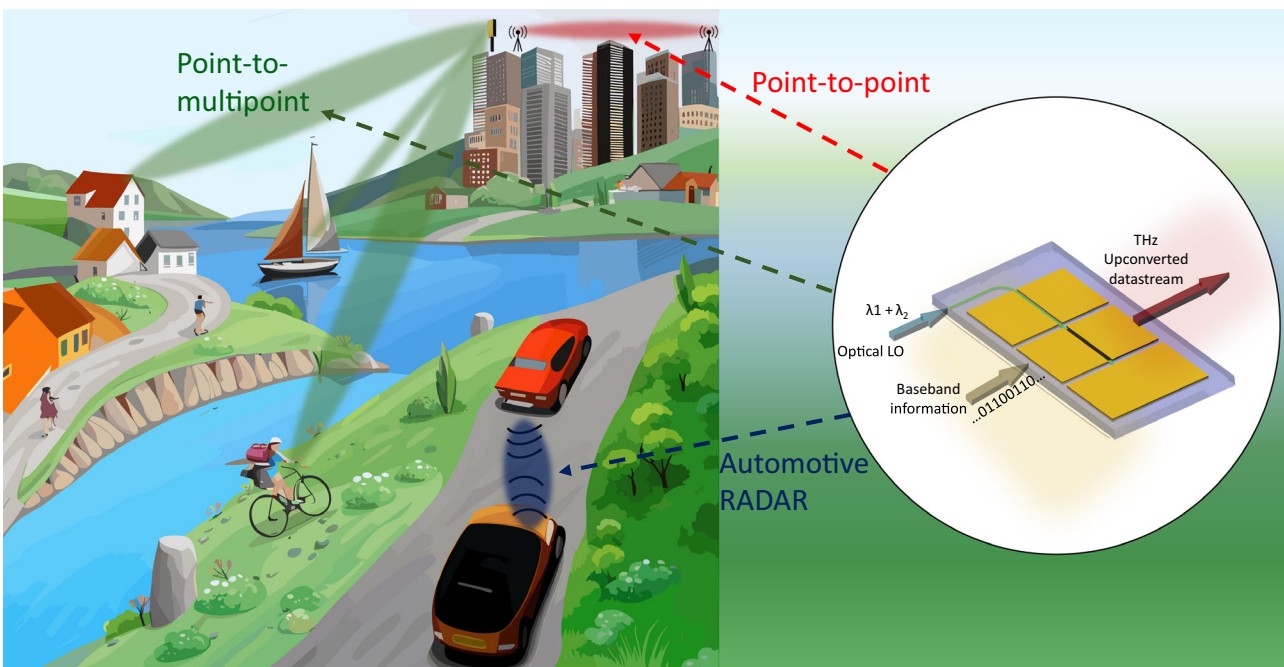

**Fig. 1 | Antenna systems for sub-THz transmission.** In these scenarios a graphene optoelectronic mixer could be used inside a transmitter to perform frequency upconversion. In red, point-to point links, allowing directional communications between two fixed network nodes. In green, point-to-multipoint links using phased-array antenna allowing to reach multi-users through beamforming, including objects in movement. In blue, automotive RADAR sensing for, e.g., autonomous driving. Each node of the network contains a frequency converter that allows the upconversion of a baseband information in the sub-THz frequency range. A graphene optoelectronic mixer (G-OEM), shown on the right, is a miniaturized object that can be potentially used in all these scenarios to perform frequency upconversion of baseband signals in the sub-THz range, in a very effective way. The device uses an optical local oscillator (LO) constituted by two optical wavelengths ($\lambda_1, \lambda_2$) instead of an electrical LO, thus performing excellent stability and easy tunablity of the sub-THz carrier frequency (CF), allowing, e.g., the realization of compact phased-array antenna systems based on optoelectronics.

red links). The second uses multiple phased-array antenna used in the wireless access networks to improve antenna gain in a chosen direction[3] (Fig. 1, green links). A further case is the sub-THz short-range RADAR imaging with increased CF allowing spatial resolution <cm, enabling high-performance autonomous driving[8] (Fig. 1, blue link). These use cases need technology development, and recent advances in photonic integration open new opportunities[9–12]. Developing the building blocks composing the transmitters and receivers of such high-capacity networks is crucial for commercial implementations. Frequency conversion is a fundamental function at both the transmitter (upconversion) and the receiver (downconversion) and is usually implemented through superheterodyne schemes[13] involving frequency mixers[14]. The approaches proposed to date for sub-THz datastream upconversion are, as discussed below, based on: (i) electronic upconverters (ii) conventional microwave photonic transmitters based on modulators and photodetectors. In the following, we compare these techniques with a third scheme, enabled by our device.

The first techniques rely on a typical scheme used to generate the sub-THz frequencies, and are based on electronic components mixing a high-frequency electrical local oscillator (LO) with a radio data signal[14], resulting in up/downconversion[14]. Electronic circuits non-idealities push the operating conditions far from theoretical figures[15] as the CF increases. The generation of high spectral purity electronic oscillators at frequencies >100 GHz is the major bottleneck limiting the performances of sub-THz systems[16], since phase noise (PN) grows with frequency. Indeed, to generate the high CF, a relatively low frequency (tens of GHz) electronic LO is upconverted, i.e., multiplied, amplified, and filtered multiple times[17,18] (Fig. 2a). This technique is power-consuming due to nonlinear signal processing and electrical signal amplifiers[19], and the resulting signal exhibits modest performance in terms of PN. A frequency multiplier increases the PN by at least 20Log(N), where N is the multiplication factor[20], and a non-negligible

contribution comes from the noise figure of cascaded amplification stages[21]. Moreover, by increasing $f \times l$ (where $f$ is the operating frequency and $l$ is the typical length of the transmission lines inside the circuit), electronic circuit design becomes critical[15], and mismatch in length and loading between differential lines can cause skew and signal degradation[15]. As CF increases, it is thus necessary to use compensation techniques for PN reduction, transmission line parasitics mitigation, and protection from spurious electromagnetic fields. As a result, the system's complexity increases[21–24].

The second scheme is based on common microwave-photonics techniques, advantageously employed to perform heterodyne mixing and overcome the issues related to a fully electronic approach[9–12]. Highly pure LO's can be obtained from two optical wavelengths spaced by the required frequency[25,26], coupled to a fast photodetector (PD). These two optical wavelengths can generate very stable electrical tones with low PN. E.g., by using optical combs, ref. 25 reports <100 dBc/Hz at 10 KHz from the 300 GHz CF and ref. 26 reports −118 dBc/Hz at 10 MHz from the 331 GHz CF. Several wireless links employing photonic transmitters have been reported in recent years (see Table 1). These schemes use optically generated LO's, that are converted from the optical to the electrical domain by making use of fast photodiodes[27]. One of the two optical wavelengths composing the optical LO is modulated with the datastream using a modulator[27] (Fig. 2b). This scheme does not require electronic upconverters. As summarized in Table 1, this is the most used technique to obtain photonic-based wireless transmitters. Despite the limited use of high-frequency electronics, the overall system complexity is only partially reduced compared to the fully electronic approach, since two discrete high-frequency optoelectronic devices are required, i.e., a modulator and a detector, often inter-connected using optical fibers (Fig. 2b), see Table 1. In this configuration, the electrical baseband datastream is first converted in the optical domain by an optical modulator and then it is converted again in the electrical domain using a

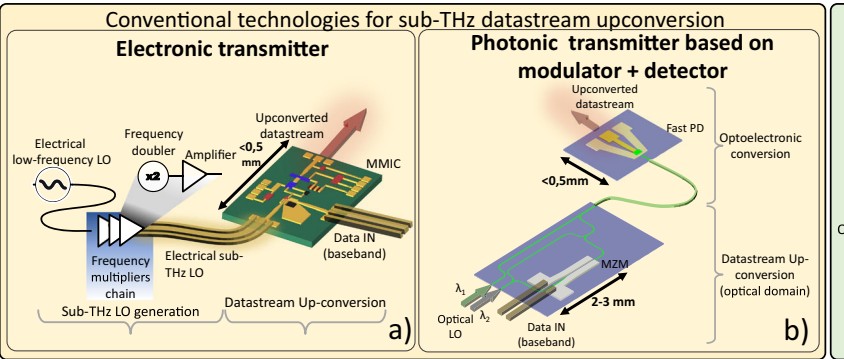

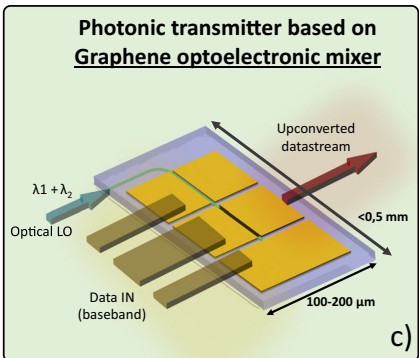

**Fig. 2 | Sub-THz transmitters technologies. a, b** Conventional implementations of sub-THz transmitters: **a** Fully electronic approach, comprising frequency multipliers for sub-THz LO generation. Once generated, the sub-THz LO is routed towards an electronic upconverter implemented using a Microwave Monolithic Integrated Circuit (MMIC) composed by electronic components, operating in the sub-THz frequency range. **b** Photonic-based transmitter, comprising an optical modulator and a fast photodetector (PD). The two components are rarely integrated into the same chip, and are most often inter-connected through optical fibers (see Table 1). **c** Our proposed implementation, comprising only one small waveguide-integrated optoelectronic mixer based on graphene (G-OEM). It allows us to reduce the complexity compared to the conventional approach shown in **a, b**, and to obtain full integration compared to the approach shown in **b**), thus enabling the realization of compact phased-array antennas based on optoelectronics.

**Table 1 | Comparison of the most recent implementations of sub-THz photonic transmitters**

| Ref. | Year | Scheme | Carrier (GHz) | Datarate[1] | Distance | Total datarate | Footprint |
|---|---|---|---|---|---|---|---|
| This work | 2022 | Int. Optoelectronic mixer | 93 | 4 Gbit/s | 2 m | 4 Gbit/s | <0,1 mm² |
| 100 | 2014 | Int. modulator + UTC-PD | 90 | 0,1 Gbit/s | 0,2 cm | 0,1 Gbit/s | ~1 mm² |
| 48 | 2018 | modulator + UTC-PD | 150 | 264 Gbit/s | 3 m | 1 Tbit/s | bulk |
| 50 | 2022 | modulator + UTC-PD | 230 | 192 Gbit/s | 115 m | 192 Gbit/s | bulk |
| 50 | 2022 | modulator + UTC-PD | 230 | 240 Gbit/s | 5 m | 240 Gbit/s | bulk |
| 27 | 2013 | modulator + UTC-PD | 237.5 | 100 Gbit/s | 40 m | 100 Gbit/s | bulk |
| 101 | 2022 | modulator + UTC-PD | 275 | 50 Gbit/s | 1 m | 50 Gbit/s | Si mod/UTC-PD[2] |
| 102 | 2020 | modulator + UTC-PD | 300 | 40 Gbit/s | 1.4 m | 40 Gbit/s | Si mod/UTC-PD[3] |
| 103 | 2018 | modulator + UTC-PD | 300 | 100 Gbit/s | 50 cm | 100 Gbit/s | bulk |
| 104 | 2020 | modulator + UTC-PD | 300 | 115 Gbit/s | 110 m | 115 Gbit/s | bulk |
| 105 | 2019 | modulator + UTC-PD | 310 | 10 Gbit/s | 58 m | 30 Gbit/s | bulk |
| 106 | 2018 | modulator + UTC-PD | 350 | 100 Gbit/s | 2 m | 100 Gbit/s | bulk |
| 49 | 2020 | modulator + UTC-PD | 380 | 300 Gbit/s | 2,8 m | 600 Gbit/s | bulk |
| 107 | 2019 | modulator + UTC-PD | 408 | 107 Gbit/s | 10 m | 107 Gbit/s | bulk |
| 108 | 2019 | modulator + UTC-PD | 450 | 132 Gbit/s | 1,8 m | 132 Gbit/s | bulk |
| 109 | 2016 | modulator + UTC-PD | 500 | 20 Gbit/s | 50 cm | 160 Gbit/s | bulk |

[1](single channel, single polarization). [2]Silicon modulator (3 mm) + bulk UTC-PD. [3]Silicon modulator (3 mm) + bulk UTC-PD.

photomixer. Each one of these two steps is characterized by conversion losses. In addition, the sub-THz antenna element is of the order of a mm², while the footprint of such transmitters is large since two discrete active devices (a modulator and a PD) are reported in most of the literature, as shown in Table 1. Instead, the next-generation phased-array antenna systems such as massive MIMOs[28] requires the use of integrated devices, and electronics is currently the only commercial solution for high-density integration[29]. As a matter of fact, most of the demonstrated photonic transmitters present in literature are based on bulk discrete systems, with few examples of integrated or partially integrated solutions (see Table 1) always exhibiting footprint >1 mm². These dimensions are still incompatible with high-density integration and preclude the realization of antenna array systems based on this approach. Indeed, one must consider a minimum amount of mm-wave electronic circuitry that would still be necessary for signal conditioning, thus the total area of the single antenna element would easily exceed the mm² limit.

We propose a third, different approach to implement a photonic transmitter, offering significant advantages in terms of power consumption, bandwidth capacity, footprint size, and PN. It is based on a self-mixing device[30], i.e., an optoelectronic mixer based on graphene (G-OEM). An OEM is a fast photomixer able to generate the high-frequency LO through optical detection, and directly mix the LO with a baseband electrical datastream[30] (see Fig. 2c and inset of Fig. 1) without the use of an optical modulator. Compared to the common photonic-aided wireless transmitters (Fig. 2b), only one G-OEM is needed to upconvert the electrical signal by internally mixing the self-generated LO with the electrical datastream[30], this last never being encoded in the optical domain during the upconversion process. Graphene has remarkable optoelectronic properties such as broadband optical absorption[31], short photocarriers lifetime[32–34] and high carrier mobility[35,36]. Graphene photonics and optoelectronics technologies are evolving toward wafer-scale production[37–39] and are compatible with the standard silicon technological platforms[37]. Optoelectronic

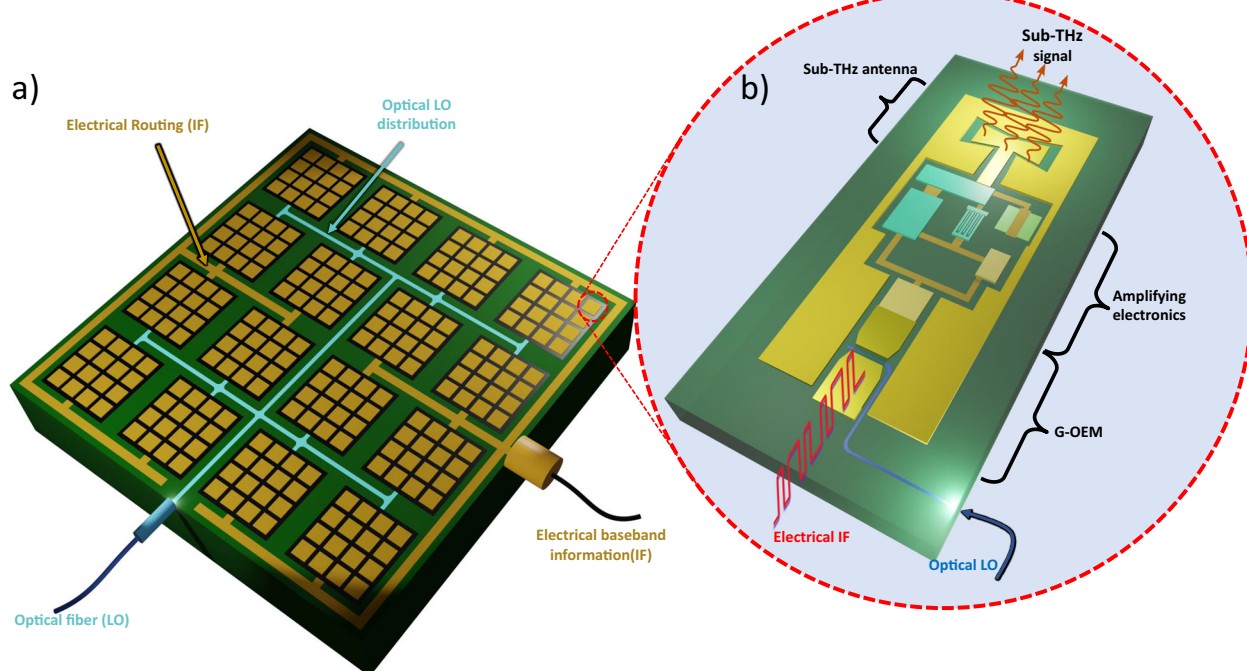

**Fig. 3 | Proposed optoelectronic antenna array system. a** The antenna is fed by an optical local oscillator and by a baseband electrical signal. A dual-wavelength optical LO is distributed inside the chip toward each element by means of an optical distribution layer. The IF signal is distributed as well towards each element using low frequency (<10 Ghz bandwidth) electrical transmission lines. No sub-THz electrical signals or sub-THz LOs are present in the antenna. **b** is a zoom of one antenna element: each element is solely constituted by a low Intermediate Frequency (IF) port, and an optical port. The sub-THz signal is generated just before being transmitted through the wireless link.

mixing using graphene PDs has been also demonstrated[40–43] and operations up to 67 GHz have been reported[42,43], but the bandwidth of G-OEMs can be much larger[44–46] as can be deduced in analogy with graphene ultrafast PDs[43–45,47]. Recent direct bandwidth measurements have shown >500 GHz bandwidth[46], according to theory predicting a 3-dB bandwidth exceeding 500 GHz[44]. An important figure of merit of upconverters is the conversion efficiency. This is defined as $P_{conv} = P_{RF}/P_{IF}$, where $P_{IF}$ is the input electrical power, while $P_{RF}$ the upconverted power. The conversion efficiency of a G-OEM depends on the charge carrier mobility[43], i.e. by the graphene quality[35,36]. Conversion efficiencies exceeding −20 dB have been predicted by theory[43]. This value, together with high-frequency operation, makes G-OEMs promising to realize sub-THz wireless links with performances potentially exceeding state-of-the-art photonic-based systems[27,48–50]. In this work we report a wireless datalink based on graphene, reaching multi-Gbit/s datarate and sub-THz CF. This result was enabled by a G-OEM used as frequency upconverter at the transmitter stage of the link. The proposed photonic transmission scheme relies on a silicon nitride (SiN) waveguide-integrated G-OEM embedded with a sub-THz coplanar waveguide (CPW). The G-OEM exhibits high conversion efficiency (−44 dB) and record (92–96 GHz) frequency operation compared to state-of-the-art G-OEMs[41–43]. The device was fabricated using hBN-encapsulated single crystal CVD graphene[36,51,52] with high carrier mobility (27000 cm$^2$ V$^{-1}$ s$^{-1}$ at room temperature). We report a 4 Gbit/s wireless link at sub-THz (93 GHz) CF, by directly mixing a baseband electrical signal (2 GHz CF) with a 91 GHz optical LO using one single device. The device reveals high potential in terms of speed, far beyond the limits of our experimental setup. The photonic upconverter footprint is <<0.1 mm$^2$, considering the active area, without access pads for on chip probing. This active area is of the same order as the active area occupied by electronic up/down converters[14,53,54], and is much smaller than any other integrated-photonic transmitter based on the scheme in Fig. 2b. The small footprint opens to application scenarios in which an integrated-photonic transmitter can be used. Figure 3 shows a

phased-array antenna system implementation approach, based on integrated G-OEMs. In the example, the antenna comprises 256 elements. Each antenna element would contain an optoelectronic mixer that converts an optically distributed LO in the electrical domain, an amplifying electronic stage, and an antenna. Phase shifting functionality may be required for beamforming/steering implementation. This is done at the IF signal level using analog or digital approaches[55,56], or at the LO level[56,57]. In the latter case, optical phase shifters need to be integrated in the antenna element[58]. The sub-THz upconversion functionality is currently realized using sub-THz electronics, comprising multi-stage mixers and amplifiers[14], schematically shown in Fig. 2a. With this conventional method, signals with CF of >100 GHz are routed to each antenna element, before being transmitted[59]. The proposed solution based on our G-OEM only requires baseband (IF) electronics (BW <10 GHz), while the upconversion is carried out by mixing the IF with an optically generated LO, routed to each antenna element using photonic waveguides instead of sub-THz electrical transmission lines[59]. The distribution of the sub-THz LO by means of electrical transmission lines can be in general power hungry since it requires the use of active sub-THz elements e.g., buffers and amplifiers[59], which are necessary to compensate for transmission lines losses and keep the SNR at acceptable levels[15,16,21,22] all along the path through the antenna elements. Using the integrated G-OEM the sub-THz signal distribution is avoided in favor of optical routing of the LO to each antenna element. Such solution drastically reduces the electronic circuit complexity and losses since the sub-THz upconverted electrical signal is generated only at the very end of the path, where an amplifying stage is present in proximity to the antenna element.

## Results
### Device design and fabrication
Figure 4a shows a schematic of the G-OEM. It consists of a CPW in ground-signal-ground (GSG) configuration, operating in the sub-THz range. The signal electrode (S) width is 74 μm and the gap between S

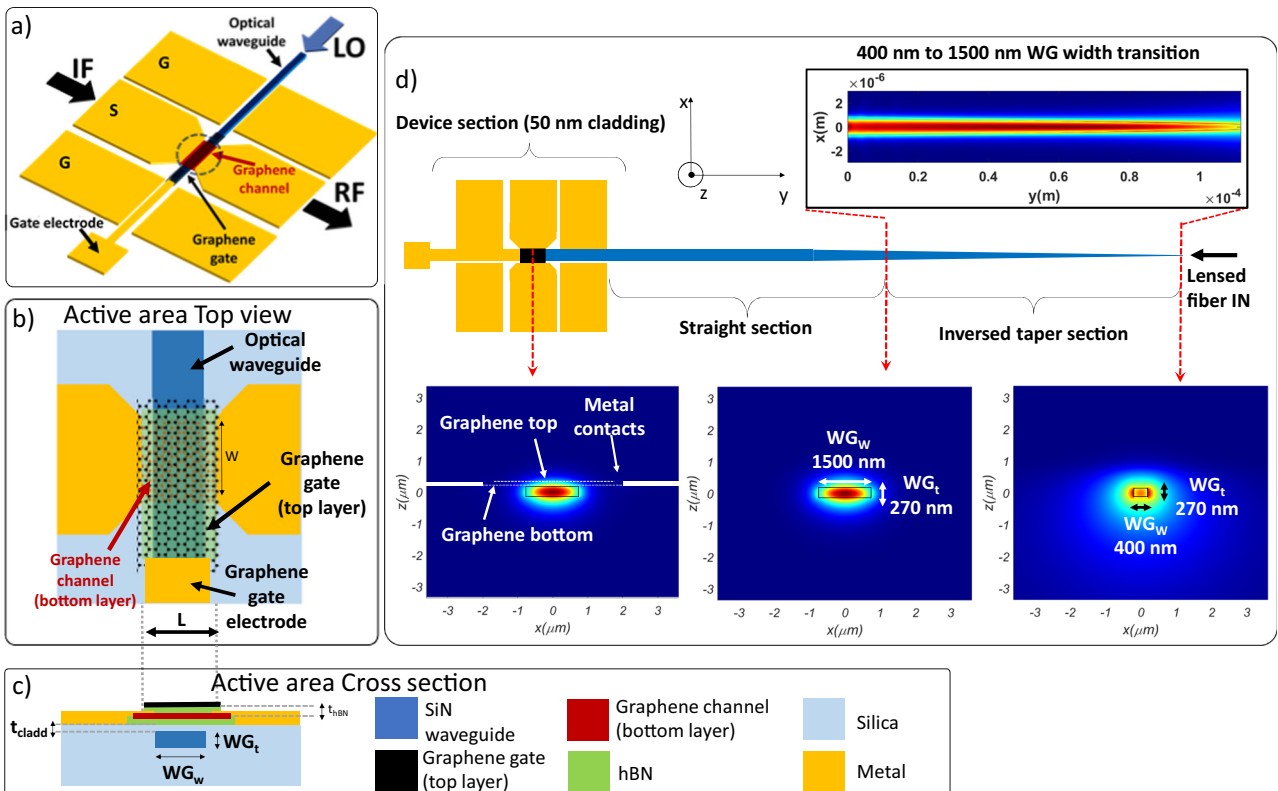

**Fig. 4 | G-OEM design. a** Schematic representation of the graphene-integrated optoelectronic upconverter. The metals constituting the electrical CPW are indicated in yellow. A graphene layer is embedded in the middle of the Signal (S) electrode of the coplanar waveguide (CPW) and is positioned on top of a SiN photonic waveguide, indicated in blue. A second graphene layer (black) acting as a top-gate is present on top of the first layer, for electrostatic doping adjustment. A baseband electrical signal at an Intermediate Frequency (IF) is fed to the device. An optical Local Oscillator (LO) is coupled to the optoelectronic upconverter by means of the photonic waveguide. The resulting upconverted Radio Frequency (RF) electrical signal is present at the output of the device. **b** Top and **c** section view of the active region, indicated with a dashed circle in **a**, comprising the photonic SiN waveguide, a graphene channel encapsulated with hexagonal Boron Nitride (hBN), and the graphene gate. $L$ is the channel length, $W$ is the channel width. WG$_t$ is the optical waveguide thickness, WG$_w$ is the optical waveguide width, $t_{cladd}$ is the thickness of the waveguide cladding and $t_{hBN}$ is the thickness of the (hBN). **d** shows the top view of the entire structure, comprising the graphene device on top of the optical waveguide with thin cladding, an inversed taper waveguide section allowing access to the photonic waveguide through butt-coupling using a lensed single-mode fiber, and a straight waveguide section with thick cladding to route the optical signal from the tapered section to the device section. For each one of the three sections, the simulated optical mode is shown.

and the ground metals (G) is 17.5 μm. This geometry provides a 50Ω characteristic impedance to match the baseband electrical signal generator and the antenna impedance (see Supplementary Information Section I.B), with negligible ohmic losses. Standard SOI technology is based on Si substrates with low resistivity, typically lower than 15 Ωcm. This is detrimental for RF performances, since ohmic losses in such substrates are non-negligible[60–62]. RF losses can be mitigated using Si substrates with high resistivity >10 KΩ cm[61]. An alternative solution is the use of a sufficiently thick oxide to avoid interaction between the electromagnetic RF propagating mode with the substrate. Our SiN technology is based on a 15 μm thick box oxide, allowing to decouple the RF mode from the Si substrate, thus minimizing ohmic losses. The electrical CPW embeds at the center of the signal electrode an hBN-encapsulated single-crystal graphene layer, transferred on top of a SiN photonic waveguide. Figure 4b, c shows a schematic detail of the active area of the device (respectively, top view and section view), indicated with a dashed circle in Fig. 4a. The graphene bottom layer (graphene channel) dimensions are $L = 4$ μm and $W = 50$ μm. A second graphene layer acts as electrostatic gate on top of the hBN-encapsulated graphene and is contacted with a metallic electrode (indicated as "Gate electrode" in Fig. 4a, b). The thickness of the hBN separating the graphene gate and the graphene channel is ~30 nm. The graphene gate is used to set the operating point of the device: the chemical potential μ$_c$ of the bottom graphene layer must be set far from the Charge Neutrality Point (CNP), i.e. μ$_c$ ~ 150 meV for optimal

operation, since at such electrostatic doping the photobolometric effect, and so the phoresponsivity, is maximized[43] (see also Supplementary Information, Section I.D-E). The hBN/graphene/hBN/graphene stack is placed on top of a quasi-transverse electric (quasi-TE) SiN photonic waveguide of 1500 nm width (WG$_w$) and 270 nm thickness (WG$_t$), allowing single mode operation at 1550 nm optical wavelength. It is worth noting that Si waveguides can be used as a valid alternative to SiN waveguides to perform OEM using graphene. Nevertheless, silicon waveguides suffer from two-photon-absorption at relatively low-power density. In the proposed perspective (optoelectronic antenna array) a relatively high input power could be coupled at the input of the chip, before being routed and split toward the single element. The SiN technology supports high optical power densities without suffering from two-photon absorption[63]. Moreover, the routing of the local oscillator toward each antenna could require a relatively long path. SiN waveguides offer ultra-low propagation loss compared to Silicon waveguides[63]. Another advantage of SiN technology is the lower thermo-optic coefficient at telecom wavelengths, making SiN photonic circuits more stable against temperature[64]. The device comprising the graphene stack and the optical waveguide ("device section" in Fig. 4d) has the optical input from one side of the photonic chip, via edge coupling. An adiabatic inversed taper ("inversed taper" in Fig. 4d) has been designed to access the photonic waveguide through butt-coupling using a lensed single mode fiber with mode field diameter ~3 μm (see Supplementary Information,

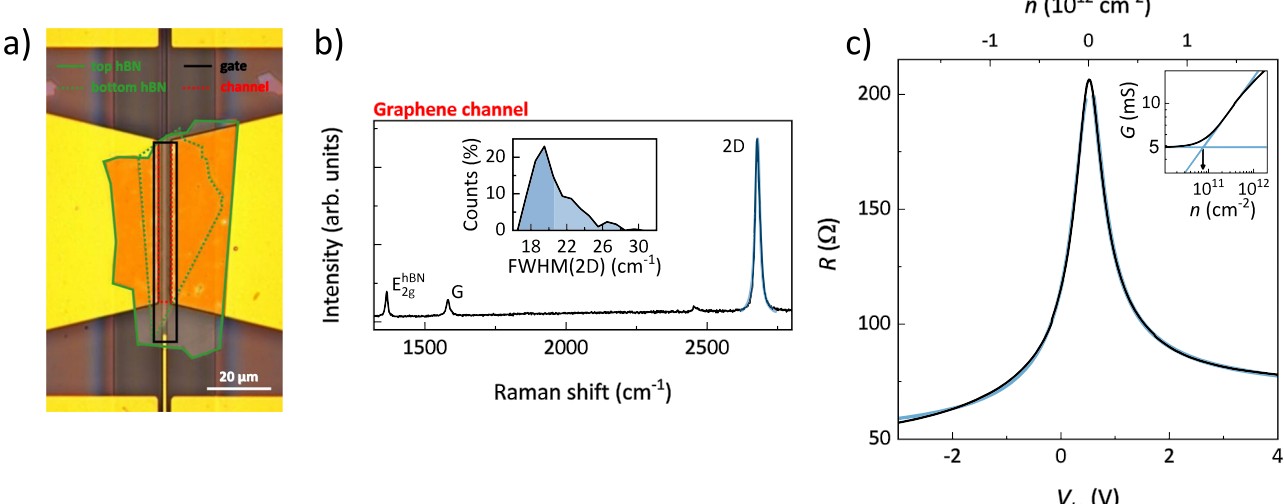

**Fig. 5 | Graphene characterization. a** Optical microscopy image of the fabricated device active region. The top and bottom hBN flakes are indicated by the continuous and dotted green lines, respectively. The graphene gate and the graphene channel are indicated by the black and red lines, respectively. **b** Representative Raman spectrum of the graphene channel (black line, acquired after the first pick-and-flip process). The light blue line is a Lorentzian fit to the 2D peak. Inset: statistical distribution of FWHM(2D) over the device channel (200 spectra in total). **c** Resistance of the device (black line) as a function of the top-gate voltage (bottom axis; the top axis shows the corresponding carrier density). The measurement is performed by applying a constant dc voltage bias of 1 mV between the source and drain contacts, while measuring the current in two-terminal configuration. The light blue lines are fitted to the resistance to extract the carrier mobility and contact resistance. Inset: log–log plot of the conductance G as a function of the carrier density, used to estimate the carrier inhomogeneity in the Charge Neutrality Point (CNP) region ($\sim 8 \times 10^{10}\,\mathrm{cm}^{-2}$).

Section I.C). The adiabatic taper gives access to a straight waveguide section ("Straight section" in Fig. 4d). The taper and the waveguide straight section are covered by a silica top cladding, with thickness of ~900 nm. The cladding is then selectively thinned down to 50 nm on top of the waveguide in the region where the hBN/graphene/hBN/graphene stack is transferred ("device section" in Fig. 4d). In this way, the optical waveguide TE mode is evanescently coupled to the graphene layer placed on top. The obtained effective absorption in the graphene active layer is ~0.057 dB/μm (see Supplementary Information, Section I.C). We used graphene as top gate to reduce the optical power losses compared to a metallic gate. From simulations, we estimated <3 dB total optical power loss due to absorption in the graphene gate and metal contacts (see Supplementary Information, Section I.C).

Figure 5a shows an optical microscopy image of the device. The full fabrication flow is presented in detail in Supplementary Fig. 11 and described in the Methods section. We use a van der Waals 'pick-and-flip' technique[51] to expose a CVD-grown graphene crystal (previously transferred from Cu to $SiO_2/Si$[36,52]) on top of a ~30 nm thick hBN flake (mechanically exfoliated from bulk crystals). After depositing Ni/Au (7/60 nm) top contacts by thermal evaporation[37], we place a second graphene/hBN stack (obtained via the same pick-and-flip procedure) onto the first one for both full encapsulation of the device channel and electrical isolation of the top graphene layer. Relatively thick hBN (>15 nm according to ref. 35) is known to crucially screen the roughness of the underlying substrate, resulting in enhanced carrier mobility in graphene[65], with optimal results upon full encapsulation in dry conditions[66].

Halfway through the assembly, we use standard scanning micro-Raman spectroscopy to evaluate the quality of the hBN-supported graphene channel. Figure 5b shows the Raman spectrum from a spatial mapping over the active area ($50 \times 4\,\mu m^2$). We observe the typical spectral features of high-quality single-layer graphene[67,68], such as a narrow Lorentzian 2D peak and large intensity ratio between the 2D and G peaks, in addition to a sharp $E_{2g}$ peak from the hBN flake. As a standard indicator for the quality of graphene/hBN stack[69], we analyze the full width at half maximum (FWHM) of the 2D peak, obtaining a statistical distribution peaked at $19.5\,cm^{-1}$ (see Fig. 5b inset). The

decisive improvement over the response of nominally identical graphene crystals placed on conventional substrates ($\sim 23\,cm^{-1}$ on $SiO_2/Si$ in ref. 37) points toward a reduction of strain fluctuations[70], which is known to crucially enhance the carrier mobility[71,72].

We measured the two-terminal resistance ($R$) of the sample as a function of the top-gate potential ($V_{tg}$), at room temperature and in air. Figure 5c shows the typical ambipolar behavior of graphene field-effect transistors, with a CNP at $V_{tg} \sim 0.5\,V$. We estimate low doping level of $n \sim 3 \times 10^{11}\,cm^{-2}$ at $V_{tg} = 0\,V$, (assuming 30 nm hBN dielectric with out-of-plane relative permittivity $\varepsilon_r = 3$). Using the standard fitting procedure from ref. 73 (light blue lines in Fig. 5c) we obtain an electron (hole) mobility of $27{,}000\,cm^2\,V^{-1}\,s^{-1}$ ($23{,}000\,cm^2\,V^{-1}\,s^{-1}$) and a contact resistance of $1.7\,k\Omega\,\mu m$ ($1.2\,k\Omega\,\mu m$). The clear asymmetry of the field-effect curve indicates a contribution from gate-dependent contact resistance[74,75], which likely affects the mobility estimate (proper four-probe measurements are however not possible in the current device design). Finally, the magnitude of charge inhomogeneity in the CNP region, estimated via the procedure in Fig. 5c inset[71], falls well below the $10^{11}\,cm^{-2}$ range, further confirming the expected high device quality[71]. Supplementary Fig. 12 described in "Methods" shows that the achievement of these results is promoted by post-fabrication thermal annealing of the device in inert atmosphere.

## Graphene OEM Operating principle and characterization

The operating principle of the G-OEM relies on the fundamental physical mechanism of hot carrier generation in a graphene layer coupled to an optical field. The chemical potential ($\mu_c$) is set far from the CNP ($\mu_c > 150$–$200\,meV$, depending on charge carrier mobility[43]). At this electrostatic doping, the dominant effect allowing photodetection is carrier heating, which translates into a decrease of the carrier mobility, i.e., into a decrease of graphene electrical conductivity[42,43]. The conductivity change can be expressed as:

$$\Delta\sigma = \sigma_{dark} + \sigma_{light} \tag{1}$$

Where $\sigma_{light}$ is the conductivity under illumination, while $\sigma_{dark}$ is the conductivity without optical excitation (dark conditions). These

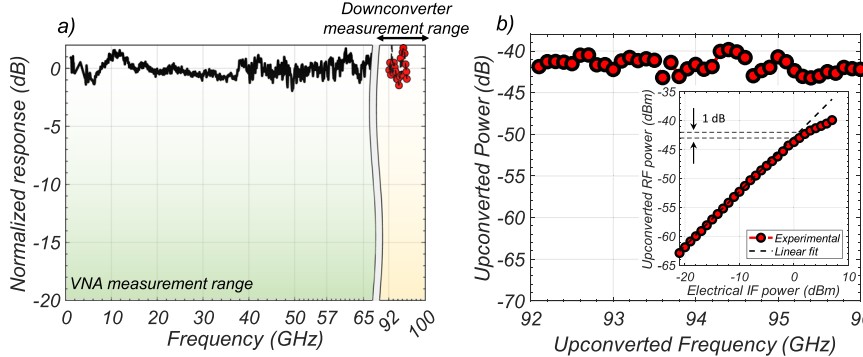

**Fig. 6 | G-OEM optoelectronic characterization. a** Frequency response of the G-OEM operated as photobolometric detector. The curve is obtained using a vector-network analyzer (VNA) up to 67 GHz, while the range 92–96 GHz is explored by means of an electronic downconverter which downconverts electrical signals from the 92–96 GHz frequency range to the 4–8 GHz frequency range. After downconversion, the signal is measured with an electrical spectrum analyzer (ESA) with a bandwidth of 44 GHz. The 3-dB bandwidth of the device is >96 GHz, since no roll-off is evidenced up to this frequency. **b** Frequency response of the device operated as OEM, specifically as an upconverter, in the range 92–96 GHz. An optical LO is used to generate a CF at 91 GHz and is mixed with an electrical sinusoidal signal (IF) with frequency tuned in the range 1–5 GHz. Thus, the upconversion lies in the 92–96 GHz frequency range. The inset of **b** shows the Upconverted power vs IF electrical input power, for IF = 3 GHz. High linearity is reached for a signal power of 1 dB, while for a 9 dBm input signal, a -3 dB compression is evidenced compared to the theoretical linear curve.

two are calculated from the Drude formula of[76,77]:

$$\sigma(T_e, \mu_c) = \frac{D(\mu_c, T_e)}{\pi(\Gamma(\mu_c, T_e))} \qquad (2)$$

Where $D$ is the Drude weight and $\Gamma$ is the transport scattering rate. Both quantities are dependent on the electron's temperature and on the chemical potential. These lasts change under optical excitation, and can be calculated from the heat equation formula:

$$-\nabla \cdot (L\sigma(T_e)T_e\nabla T_e) + \frac{C_e}{\tau_{\text{cool}}}(T_e - T_{\text{ph}}) = P(x, y) \qquad (3)$$

coupled to the charge carrier's conservation formula:

$$\frac{2}{\pi}\frac{(K_B T)^2}{(\hbar v_F)^2}\left[Li_2\left(-e^{-\frac{\mu_c}{k_B T}}\right) - Li_2\left(-e^{\frac{\mu_c}{k_B T}}\right)\right] = \frac{C_{ox}V_{GS}}{e} - \frac{\alpha\mu_c C_{ox}}{e^2} \qquad (4)$$

In Eq. (3) $L$ is the Lorenz number, $\sigma(T_e)$ is the electrical conductivity at temperature $T_e$, $C_e$ is the heat capacity $T_{\text{ph}}$ is the phonon's temperature and $P(x, y)$ is the optical power density at a specific coordinate of the graphene channel (see Supplementary Information, Section I.D for details). In Eq. 4, $v_F$ is the Fermi velocity, $Li_2$ is the dilogarithm function, $V_{GS}$ the top-gate voltage, $\alpha$ is 1 or 2 for, respectively, metallic or graphene gates(in our case, $\alpha = 2$), and $Cox = \frac{\epsilon\epsilon_{hBN}}{t_{hBN}}$ is the geometrical capacitance, which depends on the hBN dielectric constant $\varepsilon_{hBN}$ and on its thickness $t_{hBN}$ (see Supplementary Information, Section I.D for details). The photogenerated carriers' cooling dynamics are very fast in graphene, with measured relaxation times of ~2 ps[45]. This means that graphene conductivity can be modulated at very high frequencies (up to ~500 GHz) by means of an optical field. Therefore, to obtain optoelectronic mixing two optical wavelengths separated by the desired frequency (LO frequency, $f_{LO}$) are coupled to the graphene channel ("LO" in Fig. 4a). If this frequency difference ($f_{LO}$) falls in the sub-THz range, the optical LO induces a time-varying conductivity $\delta\sigma \sin(2\pi f_{LO}t)$ of the graphene channel in the sub-THz range. Thus, if an electrical sinusoid $\widetilde{V}_{in} \sin(2\pi f_{ele}t)$ ("IF" in Fig. 4a) is applied to the portion of the CPW exhibiting time-varying conductivity (i.e., where the graphene channel is present), this sinusoid is modulated at the frequency of the LO, i.e., it is upconverted. As detailed in Supplementary Information, Section I.A, this can be seen from the resulting

voltage across the device, which contains the terms:

$$\widetilde{V}_{in} \sin(2\pi f_{ele}t)\delta\sigma \sin(2\pi f_{LO}t) = \frac{\widetilde{V}_{in}\delta\sigma}{2}\cos(2\pi|f_{ele} - f_{LO}|t) + \cos(2\pi|f_{ele} + f_{LO}|t) \qquad (5)$$

In particular, the second term on the right hand of Eq. (5) contains the sum of the two frequencies, corresponding to the frequency upconversion process. The upconversion efficiency $P_{\text{conv}}$ is then used as figure of merit to evaluate the device performance. It is defined as the ratio between the output power of the upconverted tone at $f_{\text{UP}} = f_{ele} + f_{LO}$ and the power of the input electrical signal at frequency $f_{ele}$. Equation 5 comprises the particular case in which the photo-bolometer is used in its more conventional way, i.e., as a photo-detector. This corresponds in Eq.5 in substituting the time-varying electrical signal with a DC bias. Since graphene photo-bolometers operated as detectors need a DC bias, this produces a large dark current in the mA range[78]. Instead, we stress that in our case, i.e, when a graphene photo-bolometer is operated as OEM, no DC bias voltage is needed across the graphene channel[41] (see the detailed equations in Supplementary Information, Section I.A).

The fabricated OEM was first characterized in terms of photo-detection bandwidth, i.e. to determine the maximum LO frequency that can be generated by the device. As detailed in "Methods", we used two different experimental setups. The first allowed to characterize the device response up to 67 GHz using a Vector-Network-Analyzer (Keysight PNA-X 5247B). We then characterized the response in the range 92–96 GHz using a calibrated commercial electronic downconverter (Mi-Wave 970W-94/387) which downconverts high-frequency electrical signals from the 92–96 GHz range down to the 4–8 GHz range. We used a 44 GHz bandwidth electrical Spectrum Analyzer (Anritzu MS2850A) to measure the downconverted electrical signal. The result is shown in Fig. 6a and reveals a flat frequency response over the entire frequency window accessible using our measurement system. No roll-off was measured up to 96 GHz, meaning that the operating bandwidth of our device is larger than this value.

We then characterized the G-OEM as a mm-wave upconverter using the same aforementioned receiver (Mi-Wave 970W-94/387). The experimental setup is detailed in *Methods*. Figure 6b reports the upconversion of an IF signal that has been mixed with the optical LO. The IF was swept between 1 and 5 GHz, and the LO was kept at a fixed frequency of 91 GHz. The 92–96 GHz frequency window was set by the

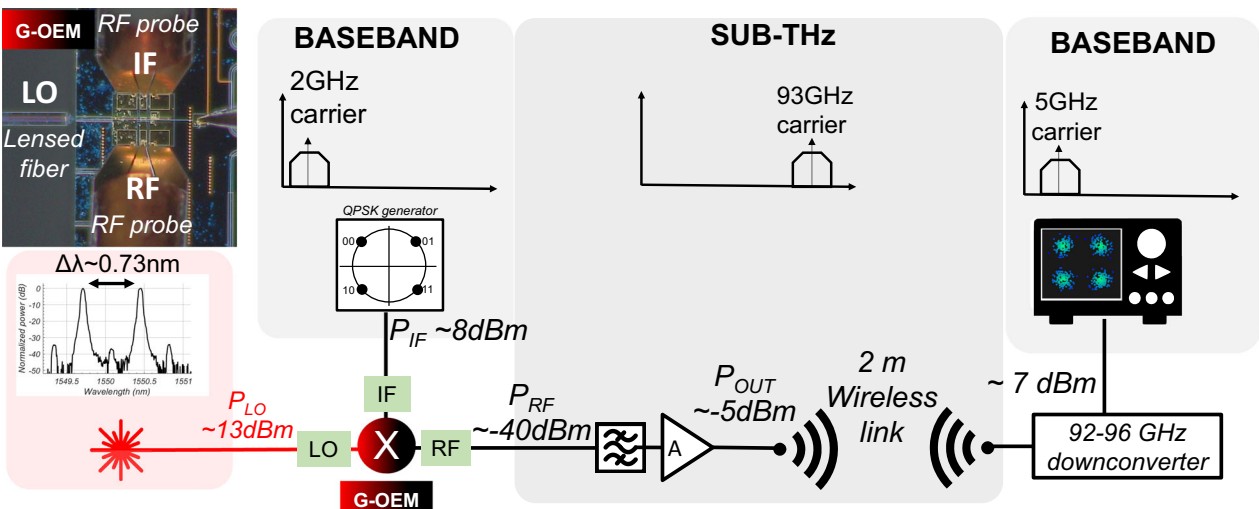

**Fig. 7 | Wireless link experimental setup.** The first input is a dual wavelength optical signal, indicated as LO. A baseband quadrature phase shift keying (QPSK) datastream (IF) with CF 2 GHz is generated using a Digital-to-Analog Converter (DAC). The data rate of the signal has been tuned between 1 Gbit/s and 4 Gbit/s. The optical and electrical inputs signals are applied to the G-OEM, as indicated in the optical image in the inset: the LO is coupled to the chip using a lensed fiber. The LO frequency is 91 GHz The IF is applied using an RF probe. The output port (RF) provides the sub-THz CF carrying the upconverted information. Since the LO frequency is 91 GHz and the IF carrier is 2 GHz, the upconverted signal is carried around a carrier frequency of 91 GHz + 2 GHz = 93 GHz (upper sideband). This electrical signal is collected by a second RF probe. After being filtered and amplified, it is transmitted through a wireless link using a horn antenna, and then is detected by a receiver composed by a second antenna and a commercial downconverter. A real-time oscilloscope is finally used to analyze the received QPSK sequence. On the bottom, the schematic electrical spectrum of the datastream is depicted, as it flows along the various elements of the system: the baseband signal is first upconverted by the G-OEM in the sub-THz domain, and then downconverted again in baseband after the wireless link, to be visualized on the oscilloscope. At each Input/Output stage of the wireless link chain, the electrical (black) or optical (red) maximum powers used during the experiment are indicated.

downconverter working range. The IF input power was 3 dBm, while the input optical power was 13 dBm. The measured upconverted power has an average of ~−41 dBm corresponding to an upconversion efficiency of $P_{conv[dB]} = P_{out[dBm]} - P_{in[dBm]} ~ -44$ dB (see Supplementary Information, Section I.E). Finally, the inset of Fig. 6b shows the characterization of the linearity of the G-OEM versus the input (IF) electrical power. The measurement was performed by sweeping the input power of a 3 GHz sinusoidal wave that was mixed with a 91 GHz, 13 dBm optical LO. The 1-dB compression point is measured for an input IF power of 1 dBm. A characterization of the conversion efficiency versus the optical LO power is also present In Supplementary Information, Section III.

## Sub-THz wireless link

Figure 7 shows the wireless transmission experimental setup. A dual wavelength laser source was realized to obtain an optical local oscillator. As detailed in "Methods", the two-phase locked wavelengths were obtained starting from a single continuous wave (CW) distributed-feedback (DBF) laser source fed into a 40 GHz bandwidth Mach-Zehnder modulator (MZM) driven by a sinusoidal signal of 45.5 GHz and operated in double sideband−suppressed carrier (DSB-SC) mode[79]. The resulting optical signal was then filtered and amplified to obtain two-phase locked optical wavelengths, separated in frequency by 91 GHz. The central carrier and high order harmonics were suppressed by >30 dB. The optical signal was coupled to the integrated graphene OEM through a lensed fiber butt coupled to the photonic chip. A quadrature phase shift keying (QPSK) baseband signal with CF of 2 GHz was generated starting from two pseudo-random binary sequences (PRBS), generated using a 65 GS/s digital-to-analog converter (DAC) (Fujitzu LEIA 55−65GSa/s 8-bit DAC), delivering a digital signal with maximum total length of $2^{17}$ samples. The datarate of the baseband signal was swept in the range 1−4 Gbit/s. The signal was fed to the G-OEM to the "IF" port indicated in Fig. 7, through a GSG RF probe (MPI T110A-GSG100) with bandwidth 110 GHz. The baseband signal was upconverted by the mixer and collected by a second probe (MPI T110A-GSG100) ("RF" port, in Fig. 7). The probe was connected through a 1-mm connector to a short (~15 cm) RF electrical cable. A 1-mm to WR-10 waveguide transition was used to connect cable to a WR-10 electrical bandpass filter (mi-wave 460W-86/94/387) with 8 GHz bandwidth and central frequency 90 GHz. The filtered signal was then amplified with an amplifier (mi-wave 955WF-35) with 35 dB gain and 75−110 GHz bandwidth, and transmitted through a 2 m wireless link using an horn lensed antenna (mi-wave 258 W) with 30 dB gain and 4 GHz bandwidth, with central frequency 94 GHz. The receiver was composed by a second antenna (mi-wave 258 W), connected to a commercial electronic downconverter (mi-wave 970W-94) allowing frequency downconversion from the 92−96 GHz (RF) range to the 4−8 GHz (IF) range. A real-time oscilloscope (Agilent Infinium VSA80000A) with electrical bandwidth of 12 GHz was used to acquire the downconverted signal and visualize the received QPSK data stream constellation using a built-in software. A picture of the setup is shown in the Supplementary Information, Section II.

To optimize the Error-Vector-Magnitude (EVM) of the received signal we adjusted the working point of the G-OEM by acting on the gate voltage ($V_G$) of the device. We used a LO power of ~13.3 dBm and an input baseband electrical power of ~9 dBm. Then, we transmitted a 1 Gbit/s QPSK baseband signal through the wireless link. As shown in Fig. 8a, a minimum EVM of 22% was found for $V_G$ ~ −1.8 V, corresponding to ~2.3 V from the CNP voltage ($V_{tg}$ ~ 0.5 V), i.e., μc-130 meV. Figure 8b, c shows the EVM as a function of the optical LO power and of the input baseband electrical power for $V_G$ ~ −1.8 V.

We then tested the EVM as a function of the datarate of the input baseband QPSK signal. Figure 9 shows the received constellation, together with the extrapolated eye diagram of one of the two quadratures (Q quadrature). The EVM for 1 Gbit/s is 22%. This value increases up to 24% for the 2Gbit/s data stream and to 27% for the 4 Gbit/s signal. We estimated[80] a bit error rate of $4 \times 10^{-6}$, $1.3 \times 10^{-5}$ and $1.3 \times 10^{-4}$ at 1 Gbit/s, 2 Gibit/s and 4 Gbit/s datarate, respectively.

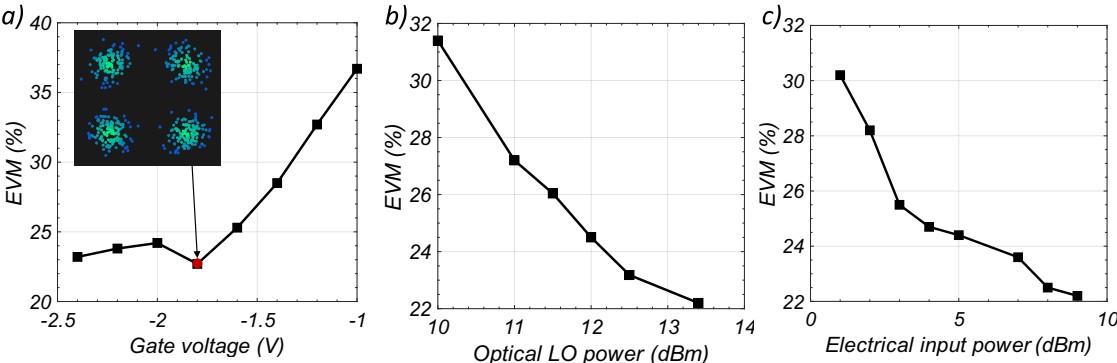

**Fig. 8 | Error-vector-magnitude (EVM) characterization. a** EVM as a function of the gate voltage applied to the G-OEM. We get an optimal working point at -−1.8 V. The corresponding measured constellation is shown in the inset. **b** EVM as a function of the optical LO power. **c** EVM as a function of the applied electrical baseband power. **b**–**c** curves are obtained for $V_G = -1.8$ V, with a 1 Gbit/s input IF datastream.

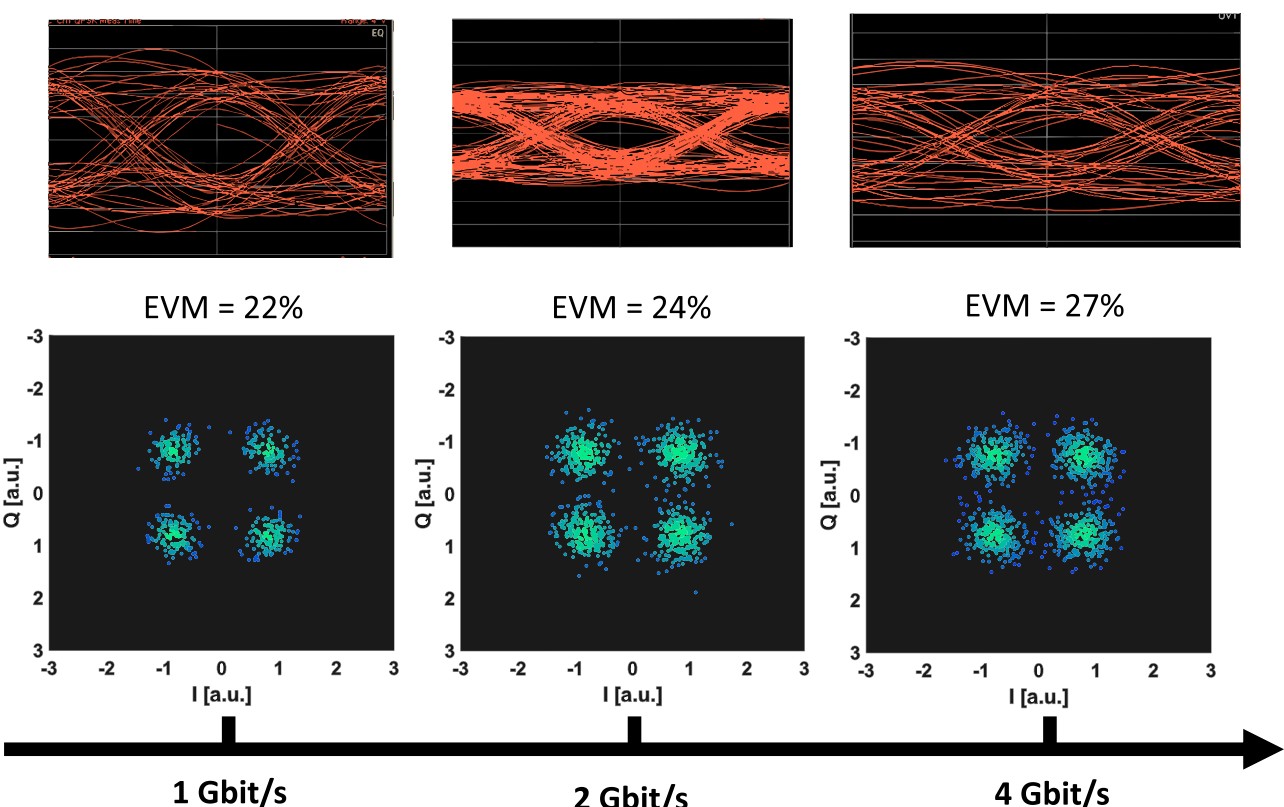

**Fig. 9 | Data transmission results.** Constellation diagram of the QPSK modulation after transmission through the sub-THz wireless link. For each datarate (from 1 to 4 Gbit/s) the eye diagram of the Q quadrature of the complex signal is shown, with the corresponding EVM. The maximum measured datarate is limited by the electrical bandwidth of the electronic components constituting the wireless link, that is fixed to 2 GHz, with central (carrier) frequency of 93 GHz.

## Discussion

The reason for the EVM deterioration as a function of datarate should be attributed to the complexity of the experimental setup that affects the signal-to-noise-ratio (SNR) of the wireless link as bandwidth increases. First, the electrical bandwidth of the full transmission chain, resulting from the electrical bandwidth of all the used components (transmission filter + antenna), is 2 GHz (with center frequency 93 GHz) which determines the maximum electrical bandwidth of the input signal (max. data rate of 4 Gbit/s). Second, the electrical signal generated by the DAC exhibits a degradation in terms of SNR while increasing the data rate, see Supplementary Information, Section IV. Consequently, the link performance is most likely setup limited. The use of antennas with higher gain would further boost these results, increasing the SNR and thus the EVM of the received signal. We used antennas with -30 dB gain, but antennas performing up to 48 dB gain at 100 GHz are commercially available (https://www.miwv.com/, https://www.eravant.com/48-dbi-gain-88-to-100-ghz-12-diameter-0-094-dia-circular-wg-w-band-gaussian-optics-antenna). This would substantially improve the performance of our wireless link, enabling longer link distances. In general, sub-THz antennas can be made more efficient by making use of on axis and off-axis elements along the path of the link[81–83]. Moreover, the performances of complex telecom radio links such as massive MIMO can be further optimized by implementing advanced techniques using, e.g., reconfigurable reflecting/

transmitting surfaces[84], and employing forward-error correction techniques (which have not been used during our experiment).

The intrinsic performances of the photonic upconverter are discussed in Supplementary Information, Section I.E, where the link between the upconversion power efficiency and the mobility of graphene is analyzed. Our simulations show that high mobility (27,000 cm$^2$ V$^{-1}$ s$^{-1}$) is beneficial compared to low mobility (<10,000 cm$^2$ V$^{-1}$ s$^{-1}$), in agreement with experimental results present in literature[41,42]. We then show that a reduction of contact resistance down to 500 Ωμm can allow a performance boost. The resulting total upconversion efficiency improvement can be at least ~11 dB, meaning that the graphene photonic upconverter can attain at least ~−33 dB conversion efficiency with small design efforts. This performance boost, together with the possibility to operate at frequencies of 500 GHz[46], would allow to reach and surpass state-of-the-art performances[5,6,11,12,18]. Indeed, the sole implementation of the discussed two improvements (use of 48 dB gain antennas and optimized mixing conversion efficiency up to −33 dB) would allow to increase the power at the receiver by >40 dB in the 2 m wireless link, which would drastically improve the SNR allowing higher modulation formats (i.e., higher channel capacity), or equivalently enable >100 m wireless links, keeping the same performances reported in the present work[15] in terms of datarate.

We now briefly quantitatively discuss the proposed photonic-aided antenna array presented in Fig. 3. The current electronic antenna array transmitters operating at ~100 GHz carrier frequency require a power consumption in the range 1–10 W and deliver an output power in the range 10–15 dbm[85–87]. Each antenna element based on our proposed scheme contains a G-OEM providing ~−40 dBm. To attain total output power of 15 dBm, a 30 dB amplifying stage is needed for each element. Each amplifier should require few tens of mW power to keep the power consumption within some W. Available electronics[88] allows ~20 mW power consumption for each element, to obtain ~30 dB power amplification. Concerning the optoelectronic mixer, it should require negligible power compared to the amplifying stage (i.e., <<10 mW for IF and LO signals). The G-OEM power consumption can be improved acting on the mobility of the graphene detector. This is quantitatively discussed in Supplementary Information, Section I.E. Also, high photoresponsivity enhancement can be achieved using plasmonic detectors and/or stacking more than one graphene layer. ref. 89 reported 0.5 A/W external responsivity with <0.6 V applied to the detector contacts, and compact size (<6 μm$^2$ active region). These devices could be advantageously used in antenna arrays. Indeed, the detector presented in[89] provides a responsivity allowing comparable performances as the one presented in our work in terms of conversion efficiency, but thanks to plasmonic enhancement it requires only an optical LO input power <0 dBm and an IF input power ~0 dBm if used as G-OEM, thus matching the power consumption requirements of the single antenna element. Finally, an antenna array with such characteristics should preserve high gain to keep the same performances as the one presented in our work, in terms of link distance and data rate. At this regard, 40 dB gain have been already demonstrated at carrier frequencies >100 GHz for 16 × 16 antenna arrays[90].

Finally, let us comment on the perspective for scalability of the proposed G-OEM. We stress that the CVD graphene employed as channel and gate material is obtained via growth and transfer methods with established wafer-scale capabilities[37]. The carrier mobility necessary for effective G-OEM operation (>10$^4$ cm$^2$ V$^{-1}$ s$^{-1}$ at room temperature), crucially requires decoupling of the graphene channel from extrinsic scattering sources[91]. As demonstrated both for exfoliated graphene flakes[65,66], and CVD-grown graphene single crystals[92,93], screening of substrate roughness is paramount. The only reliable strategy currently available to achieve this requires the use of thick hBN[35]. Although large-scale growth of single or few-layer single-crystalline hBN is well established, the few-layer thickness limit does not

guarantee adequate screening, resulting in carrier mobility in line to that of graphene on SiO$_2$ (<10$^4$ cm$^2$ V$^{-1}$ s$^{-1}$)[94]. The synthesis of scalable counterparts of thick hBN flakes (i.e., providing an adequate screening effect) is a sensitive topic. Relevant advances in crystals dimensions can be found in ref. 95, where a 2 × 5 cm$^2$ trilayer hBN crystals growth has been demonstrated. Similarly, the growth of large area (5 cm lateral dimension) thick (5 nm) hBN have been shown in[96], with graphene mobility values approaching ~10$^4$ cm$^2$ V$^{-1}$ s$^{-1}$. We showed that the presented G-OEM fabrication protocol is compatible with CVD-grown materials (in this case, graphene itself), making it ready for the integration of large-scale thick hBN films.

In summary, we have reported a wireless link based on graphene, using a hybrid electronic/photonic approach based on an integrated device, the G-OEM. This transmitter targets radio applications in the sub-THz band to improve performance parameters such as frequency accuracy, PN, upconversion efficiency, and to envisage an integrated solution that fits in the sub-THz antenna element footprint. The realization of sub-THz radio links enables telecom operators to achieve network densification while optimizing overall network capacity and latency performance (e.g., deploy more links per square km with small and picocells or new cell-free network topologies[28]). Approaching 100 GHz CF, electronic technology is showing its limits, while the common photonic implementation is still far from allowing the realization of single antenna elements. The present work assesses the merit of the realization of an integrated optoelectronic sub-THz upconverter suitable for single or multiple antenna systems, opening new perspectives for the use of integrated photonics in next-generation antenna arrays in 5 G and 6 G NR, potentially solving the issues related to the concurrent technologies. We show setup limited record frequency operation (93 GHz) and datarate (4 Gbit/s), using high-quality CVD graphene allowing high values of upconversion efficiency (~−44 dB) compared to state-of-the-art G-OEMs[42,43,97]. Eventually, we describe the pathways towards a substantial increment of both datarate and conversion efficiency by acting on the design at both the system and the device level. We use our device as a photonic upconverter, nevertheless it can be used as downconverter as well[39,41–43], thus allowing the realization of the receiver using the same graphene technology, in view of the realization of a G-OEM-based full-link. This work opens the route to compact, low-power, cost-effective sub-THz antenna array systems fully based on graphene as active material, with performances potentially overcoming state-of-the-art limitations[5,6,11,12,18].

# Methods
## Device fabrication
The complete flow developed for the fabrication of the G-OEM is depicted in Supplementary Fig. 11, with steps numbered from 1 to 10. Central to our process is a modified version of the pick-and-flip method described in ref. 51. This technique consists of picking up hBN and graphene, respectively, with a polymeric stamp, and then depositing it face down (i.e., with the hBN flake on top) on a second stamp, to obtain graphene exposed on top of hBN when the stack is finally released on the target substrate. Two polymer membranes—poly(bisphenol A carbonate)polycarbonate (PC) and poly(vinyl alcohol) (PVA) —are deposited on a polydimethylsiloxane (PDMS) stamp, which is attached to a glass slide. With the aid of a micromechanical stage, this structure is used to pick-up an hBN flake and then a portion of a graphene crystal from SiO2 (steps 1 and 2). Subsequently, the resulting stack is put in contact with a second underlying PDMS stamp (step 3). To transfer the graphene/hBN from the first to the second stamp, the solubility of PVA in water is exploited. Indeed, when a drop of water is released between the two stamps, the PVA is dissolved, and the first stamp is detached from graphene/hBN (step 4). Subsequently, the second stamp is flipped and aligned to the SiN photonic waveguide. The hBN/graphene stack is thus deposited, leaving the graphene exposed (step 5). The sample is heated on the assembly stage to

improve the adhesion to the substrate (step 6). The graphene is patterned by electron beam lithography followed by oxygen plasma etching into a $50 \times 8\,\mu m^2$ rectangle. Two contacts are fabricated on top of graphene (with $2\,\mu m$ superposition on each side, resulting in a final channel of $50 \times 4\,\mu m^2$) using electron beam lithography followed by thermal evaporation of Ni/Au and lift-off (step 7). To fabricate the graphene gate, the pick-and-flip technique is used again (steps 1–4), and a second hBN/graphene stack is deposited on the device (steps 8 and 9). The top graphene is finally contacted with the gate electrode (step 10). By doing so, the top hBN flake has the double function of encapsulant for graphene and of top-gate dielectric.

Immediately after fabrication, the DC transport properties of the G-OEM are comparable to those of CVD graphene single crystals on SiO₂ (dotted black line in Supplementary Fig. 12)[37], indicating poor quality of the graphen/hBN interfaces. We attribute this to the sequential stacking method for encapsulation, which traps polymeric residuals from processing the top contacts. We find that annealing the structure in inert atmosphere results in a dramatic increase of the carrier mobility (continuous black line in Supplementary Fig. 12), likely due to the aggregation of interface contaminants[98].

### Optoelectronic frequency response measurement

The optoelectronic frequency response of the G-EOM allows to evaluate of the optical LO frequency detectable by the device, and corresponds to the optoelectronic bandwidth of the device when used as photodetector. The characterization was performed using two experimental setups, covering two different frequency windows. The scheme in Supplementary Fig. 13 shows the first setup, while the two images show the elements indicated in the scheme. The setup consisted of a Vector-Network-Analyzer (VNA) (Keysight PNA-X 5247B), connected to a MZM, used to modulate a $1.55\,\mu m$ wavelength CW distributed-feedback (DBF) laser source. The modulating frequency was swept up to 67 GHz. We accounted for the MZM 40 GHz electro-optical bandwidth by measuring its response during the VNA calibration, using a 70 GHz bandwidth photodiode (Finisar XPD3120R) with known frequency response. The modulated optical signal was then coupled to the G-OEM using a tapered optical fiber. The photobolo-metric response characterization requires a DC voltage bias[99] applied to the graphene channel. This was fed using two bias tees connected at each side of the G-OEM. A DC needle was used to contact the gate pad and set the optimal graphene chemical potential $\mu_c$ optimizing the optoelectronic response. In Fig. 8a we show an optimal response for $V_G - V_{CNP} = 2.3\,V$. The G-OEM was contacted with RF GSG probes (MPI T110A-GSG100). One side of the device was AC coupled to ground, while the other side was connected to the second port of the VNA. The electrical power resulting from the photocurrent flowing through the $50\,\Omega$ load of the VNA was measured as a function of the electrical frequency driving the MZM. The result of the characterization is shown in Fig. 6a, and reveals a flat response over the entire frequency range.

Because of the frequency limitation of the VNA, we used a different setup to characterize the optoelectronic bandwidth of the G-EOM in the frequency range used for the wireless transmission experiment. First, we implemented a phase-locked dual-wavelength tunable laser source shown in Supplementary Fig. 14. We used one port of the VNA as signal generator, which was manually swept in the range $f_0 = 46$–48 GHz. We operated the MZM in double sideband–suppressed carrier (DSB-SC) mode[79], to obtain two optical wavelengths separated by $2f_0$. To further suppress the central wavelength, we used a band-stop narrow fiber grating filter centered at $1.55\,\mu m$, with optical bandwidth of -0.241 nm(-30 GHz). The resulting signal was then amplified and filtered again with a 1 nm (-125 GHz) optical pass-band filter to reduce the amplified spontaneous emission (ASE) power of the optical amplifier. The resulting spectrum of the optical signal obtained with an optical spectrum analyzer (OSA) is shown in Supplementary Fig. 14. We calibrated the modulation depth of the MZM as a function of the

modulating frequency by measuring the optical power of the two generated wavelengths on the OSA compared to the central wavelength and higher order harmonics. We measured suppression of both carrier and higher-order harmonics of >30 dB, as shown in Supplementary Fig. 14.

The two wavelengths were then coupled to the G-OEM to measure the optoelectronic frequency response of the device in the range around the wireless link experiment CF. The setup is shown in Supplementary Fig. 15. The beating of the two wavelengths on the device produced a photocurrent in the range $2f_0 = 92$–96 GHz. The G-OEM was then connected to an electronic downconverter (Mi-Wave 970W-94/387), which downconverted the sub-THz signal in the range 4–8 GHz. The input RF port of the downconverter had a WR-10 waveguide connection. A 1-mm coaxial to WR-10 transition was used to connect the RF probes to the electronic component, as shown in the image of Supplementary Fig. 15. The downconverted electrical signal was visualized on an electrical spectrum analyzer (ESA) (Anritzu MS2850A). We corrected the amplitude of the measured signal accounting for the downconverter frequency response, which oscillated by -2 dB in the considered frequency range, and for the loss of the RF cable connected between the downconverter and the ESA. The frequency response is shown in Fig. 6a (red dots) and does not present any roll-off in the considered range.

### Conversion efficiency measurement

The setup used to characterize the G-OEM conversion efficiency is shown in Supplementary Fig. 16. It is very similar to the one used to characterize the optoelectronic response in the range 92–96 GHz. The same dual-wavelength laser source described above and shown in Supplementary Fig. 14 has been used, without sweeping the modulating frequency of the MZI. Specifically, we fixed $f_{LO} = 91\,GHz$. When the device is used as optoelectronic mixer, there is no need to apply any DC bias voltage along the graphene channel as can be deduced from equations S4-S10 in the Supplementary Information. Thus, the DC voltage supply used for the "Optoelectronic frequency response measurement" is essentially substituted by an AC signal generator, representing the IF frequency fed to the mixer. The RF output carrying the upconverted frequency was then connected to the same electronic downconverter presented above. The upconverted frequency was in the range $f_{RF} = f_{IF} + f_{LO} = [92-96]\,GHz$, and so in the range 4–8 GHz after downconversion. The signal was visualized on a spectrum analyzer.

## Data availability

The Source Data underlying the figures of this study are available at https://doi.org/10.5281/zenodo.8341503. All raw data generated during the current study are available from the corresponding authors upon request.

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

## Acknowledgements

This work was supported by the European Union's Horizon 2020 research and innovation program, through GrapheneCore3 under grant 881603. K.W. and T.T. acknowledge support from the JSPS KAKENHI (Grant Numbers 21H05233 and 23H02052) and World Premier International Research Center Initiative (WPI), MEXT, Japan.

## Author contributions

A.M., V.S., M.R. carried out the device simulation and design. G.P., V.M., M.G., S.P. fabricated the device. G.P., V.M., S.P. characterized the graphene samples. S.S. fabricated the passive photonic circuit. K.W., T.T. grew the hexagonal boron nitride crystals. A.M., M.R., V.S. conceived the experiment. A.M., M.R, V.S., A.D., L.G carried out the experimental work. M.R., C.C., S.P., A.D. supervised the work. A.M. wrote the paper, with contributions from all authors.

## Competing interests

The authors declare no competing interests.
