## [Peer Review File · Nature Communications]

Sub-THz wireless transmission based on graphene integrated optoelectronic mixerREVIEWER COMMENTS

Reviewer #1 (Remarks to the Author):

Review "Sub-THz wireless transmission based on graphene integrated optoelectronic mixer "
The manuscript "Sub-THz wireless transmission based on graphene integrated optoelectronic mixer "
is well written and deals with an interesting approach combining graphene and wireless datalink at frequencies around 96 GHz.

The authors claim to overcome conventional technologies such as standard electronic transmitters and standard photonic transmitter which consist of modulator and detectors which are rarely found on the same chip. The proposed graphene optoelectronic mixer is a compact device where mixing and detection can be realized on the same chip.

Optoelectronic mixers have been demonstrated at high frequencies already. The conversion efficiency of a graphene optoelectronic mixer is defined by the charge carrier mobility which is a characteristic value for the graphene quality. In order to achieve high values of graphene mobility as shown in the manuscript the CVD grown graphene needs to be encapsulated in hBN. The authors report a 4 Gbit/s wireless link at 93 GHz carrier frequency. They used the van der Waals pick and flip technique to combine a CVD grown graphene x-tal and the 30 nm thick hBN flakes. This procedure is used twice in order to obtain two graphene layers with high carrier mobility.

The authors do not mention that the scaling perspective of graphene encapsulated in hBN flakes is not comprehensible. For this reason, I do not see that the proposed devices enable a totally new field of application.

I do see that there is advancement compared to the publication of some of the authors in ACS Photonics 2021, 8, 369–375 with the approach and the results shown in the current manuscript. The measurements of the data link are in this way new for this kind of device.

In my opinion the shown results are interesting for everyone in the field working with data transmission links. Moreover, I do think that a critical analysis of the manufacturability of devices with high graphene quality should be discussed. The shown speed and device performance is nothing outstanding for anyone familiar with graphene photodetectors. Nevertheless, the combination of a high speed graphene device in such a communication link is new to my knowledge and thus worth to publish.

Small error in figure caption: fig. 4. C) "active area cross section"

Reviewer #2 (Remarks to the Author):

In this paper, the authors report a sub-THz wireless transmission based on graphene integrated optoelectronic mixer, where a bandwidth >96GHz and -44 dB upconversion efficiency are achieved, paving the way to the development of novel arrayed-antennas for millimeter-wave technology relying on a new approach taking advantage of optics.

Therefore, the paper will be of interest to the community and I recommend this article to be published in Nature Communications after answering the following questions:

1. The bottom graphene is encapsulated by hBN. The functionality of the hBN need to be clarified. Moreover, since the hBN is mechanically exfoliated from bulk crystals and cannot be easily repeated in a mount of samples, why not use a CVD grown dielectric material?
2. The material of the G-OEM waveguide is SiN, and it can be easily fabricated with Si waveguide based on SOI platform. The authors need to explain the benefit of SiN waveguide in both performance and cost.
3. On top of the Fig 6., the sentence is not complete "... Finally, the inset in ...". The authors need to modify it.
4. In the fabrication method section, the authors should state the details of graphene geometry made.
5. The equivalent circuit diagram of the G-OEM should be clarified. Also, the authors should theoretically discuss and calculate the bandwidth limit of the system. An expectation of how to improve the system is also needed since the graphene experimental measurements revealed a response time of 2.1ps.
6. As the G-OEM is promising to be used in arrayed-antennas, the fabrication method in this article is rather complicated. The authors could discuss how to push this G-OEM to mass-production.

Reviewer #3 (Remarks to the Author):

A manuscript by A.Montanaro with co-authors discusses a 92-96 GHz wireless data link, where the focus is made on electro-optical mixer made of electrostatically gated graphene.

The manuscript is very interesting and authors have obviously made novel, large and interesting experimental work. The authors use a novel approach of up-conversion of the baseband signal (1-4GHz) into a sub-THz spectral range (here, 92-96GHz) utilized an LO generated in the same graphene channel as a beating of two lasing lines in the 1.55 μ m wavelength range. This way the authors manage to avoid any additional photonic or THz modulators, hence reducing the device complexity significantly. It is very interesting to see an application demonstration in a sub-THz data link in a room scale (2 meters). A presentation of the graphene electro-optical mixer (G-EOM) up-conversion efficiency based on internal graphene parameters was also done (see some of my remarks).

I find this work sufficiently novel, and interesting.

With an overall positive view on this manuscript I have some questions, answering which would, in my opinion, improve the quality of the work. I sort my questions based on the category rather than on sections in the manuscript.

1. Authors bring to our attention possible improvements of the G-EOM from the current stage. E.g. an up-conversion efficiency increase by 11dB (see e.g. Discussion). However, based on page 8 in Supplementary Materials, the aforementioned 11dB improvement is justified is the contact resistance to the graphene channel is reduced (e.g. down to 500 μ m). Well, on page 9 in SM, authors present performance simulation if the carrier mobility reduces from the current 27000 down to a hypothetical 4000, observing a 15dB efficiency reduction. However, we do not know what can we expect from a, possible, mobility increase above 27000.
2. Related to the previous question. For a reduction of the contact resistance to the graphene channel (down to 500 μ m) authors give a reference [18], authored by basically the same team. My question is

the following, why, holding the technology for a lower R_c , why this technology was not used in the current work?

3. A large emphasize is placed on an array of the discussed devices, Fig.3 a discussion around it. On the other hand, both layers of graphene (the channel and the top-gate) as well as hBN were made by exfoliation. I think authors have to present possible scenarios for the device fabrication which are compatible with the definition "wafer-scale).

4. Authors present diverse experimental set-ups for studying different aspects of the G-EOM. However, I find it extremely important to show experimentally what effect the LO power (hence the laser power) will have on the up-conversion efficiency. Specifically since simulations are done for a 20mW laser power and the measurements for a -13dBm. Interestingly, EVM data are given for a range of the laser power (Fig.8), but not for the efficiency.

5. I do not find a discussion for of the W-band horn antenna gains as a way to improve link range particularly well done. It is a common knowledge that antenna-to-antenna signal transfer is a function of the antenna gains. But it is seems that authors forget that utilization of on-axis (lenses) and off-axis (mirrors) optical elements can easily make a sub-THz link very efficient, at least on distances over 100-500m. I do not say that authors are wrong with their discussion, but they can make this discussion more professional.

6. Discussion over the "foot-print" of this device vs other devices. Indeed, in most of areas of communication the device scale, their integration possibilities, have not only effect on the system compactness/price but also on functionality extension, e.g. by making arrays, etc. However, I am skeptical in taking just the active graphene area in the consideration. Despite of possible ways of integration, G-EOM-based link is still a complex device, including the lasers, filters, amplifiers, antennas, etc. Do authors have good arguments how to package (ok, excluding the lasers) into the 0.1mm patch?

7. Line 413, instead of Fig.8 it is probably Fig.6?

8. On many occasions letters in figures are way too small to be seen in a printed version.

Answer to reviewers

Reviewer #1

Remark n.1

Review “Sub-THz wireless transmission based on graphene integrated optoelectronic mixer “
The manuscript “Sub-THz wireless transmission based on graphene integrated optoelectronic mixer “ is well written and deals with an interesting approach combining graphene and wireless datalink at frequencies around 96 GHz.

The authors claim to overcome conventional technologies such as standard electronic transmitters and standard photonic transmitter which consist of modulator and detectors which are rarely found on the same chip. The proposed graphene optoelectronic mixer is a compact device where mixing and detection can be realized on the same chip.

Optoelectronic mixers have been demonstrated at high frequencies already. The conversion efficiency of a graphene optoelectronic mixer is defined by the charge carrier mobility which. Is a characteristic value for the graphene quality. In order to achieve high values of graphene mobility as shown in the manuscript the CVD grown graphene needs to be encapsulated in hBN. The authors report a 4 Gbit/s wireless link at 93 GHz carrier frequency. They used the van der Waals pick and flip technique to combine a CVD grown graphene crystal and the 30 nm thick hBN flakes. This procedure is used twice in order to obtain two graphene layers with high carrier mobility. The authors do not mention that the scaling perspective of graphene encapsulated in hBN flakes is not comprehensible. For this reason, I do not see that the proposed devices enable a totally new field of application.

Answer n.1:

We thank the Reviewer for the comment. It is well established that coupling of graphene and hBN [Ref. 65 (*new version of the manuscript*)] is the strategy of choice for high carrier mobility at room temperature ($>10^4 \text{ cm}^2\text{V}^{-1}\text{s}^{-1}$). This results from screening of substrate-induced disorder [Ref. 93 (*new version of the manuscript*)], which is however effective only when relatively thick hBN is employed ($>15 \text{ nm}$ according to Ref. 35 (*old and new version of the manuscript*)). The synthesis of scalable counterparts of the hBN flakes (i.e., providing an adequate screening effect) is a sensitive topic. Relevant advances in crystals dimensions can be found in Ref. 97 (*new version of the manuscript*), where a $2 \times 5 \text{ cm}^2$ trilayer hBN crystals growth has been demonstrated. The authors of Ref. 97 (*new version of the manuscript*) claim that larger area hBN crystals growth is conceptually possible. Regarding thicker crystals, a remarkable result has been reported in Ref. 98 (*new version of the manuscript*), where the synthesis of large area (5 cm lateral dimension) 5 nm hBN is reported. Concerning the graphene layers, we stress that the two layers of graphene acting as active channel and as gate are CVD grown, and they are transferred using wafer-scale techniques [Ref. 37 (*old and new version of the manuscript*)]. As such, we believe that we do take a first important step toward scalability. In addition, by using CVD graphene, we demonstrate that the van der Waals assembly techniques used in the manuscript are suitable for CVD-grown materials. The developed G-OEMs are therefore ready to integrate all-CVD stacks: this will be done as soon as this approach becomes appropriate performance-wise.

In the manuscript, we added the following discussion paragraph regarding scalability:

“Finally, let us comment on the perspective for scalability of the proposed G-OEM. We stress that the CVD graphene employed as channel and gate material is obtained via growth and transfer methods with established wafer-scale capabilities³⁷. The carrier mobility necessary for effective G-OEM operation ($>10^4 \text{ cm}^2\text{V}^{-1}\text{s}^{-1}$ at room temperature), crucially requires decoupling of the graphene channel from extrinsic scattering sources⁹³. As demonstrated both for exfoliated graphene flakes^{65,66}, and CVD-grown graphene single crystals^{94,95}, screening of substrate roughness is paramount. The only reliable strategy currently available to achieve this requires the use of thick hBN³⁵. Although large-scale growth of single or few layer single-crystalline hBN is well established, the few-layer thickness limit does not guarantee adequate screening, resulting in carrier mobility in line to that of graphene on SiO_2 ($<10^4 \text{ cm}^2\text{V}^{-1}\text{s}^{-1}$)⁹⁶. The synthesis of scalable counterparts of thick hBN flakes (i.e., providing an adequate screening effect) is a sensitive topic. Relevant advances in crystals dimensions can be found in⁹⁷, where a $2 \times 5 \text{ cm}^2$ trilayer hBN crystals growth has been demonstrated. Similarly, the growth of large area (5 cm lateral dimension) thick (5 nm) hBN have been shown in⁹⁸, with graphene mobility values approaching

$\sim 10^4 \text{ cm}^2\text{V}^{-1}\text{s}^{-1}$. We showed that the presented G-OEM fabrication protocol is compatible with CVD-grown materials (in this case, graphene itself), making it ready for the integration of large-scale thick hBN films.”

Remark n.2

I do see that there is advancement compared to the publication of some of the authors in ACS Photonics 2021, 8, 369–375 with the approach and the results shown in the current manuscript. The measurements of the data link are in this way new for this kind of device. In my opinion the shown results are interesting for everyone in the field working with data transmission links.

Answer n.2:

We thank the Reviewer for the positive opinion on our manuscript and for acknowledging the advancement over previous works. Compared to Ref. 35 (*old and new version of the manuscript*), our work complements the G-OEM model, more specifically the circuit level model. Moreover, this work demonstrates the first integrated photonic G-OEM. Above all, we stress that we do not merely report the characterization of a new device: we embed it in an application-relevant testbed, demonstrating the first wireless transmission based on graphene. Our experiment also opens a perspective on the realization of conceptually new systems based on optoelectronic antenna arrays.

Remark n.3

Moreover, I do think that a critical analysis of the manufacturability of devices with high graphene quality should be discussed.

Answer n.3:

We thank the reviewer for this clarification request. CVD graphene single crystals like the ones used in our work are defect-free (see Ref. 51 of the *old version of the manuscript*, now Ref. 52), meaning that the carrier mobility is limited by extrinsic disorder [Ref. 93 (*new version of the manuscript*)], induced by the substrate and/or the transfer process. While over the years the latter has been largely optimized – with wafer-scale compatible protocols currently available [Refs. 36-37 (*old and new versions of the manuscript*)] – no substantial progress has been made in establishing dielectric substrates as effective alternatives to hBN. Thick (>15nm) hBN remains the only dielectric environment guaranteeing a mobility of $>10^4 \text{ cm}^2\text{V}^{-1}\text{s}^{-1}$ according to Ref. 35 (*old and new version of the manuscript*). The development of techniques for the synthesis of large-scale hBN is a sensitive topic, but it goes beyond the scope of our current work and should be rather addressed at the community level. Nonetheless, we added in the manuscript the references to relevant, recent advances on the synthesis of scalable counterparts of thick hBN flakes (i.e., providing an adequate screening effect). These are Ref. 97 (*new version of the manuscript*), which shows a $2\times 5\text{cm}^2$ trilayer hBN crystals growth, and Ref. 98 (*new version of the manuscript*), showing the growth of large area (5 cm lateral dimension) thick (5 nm) hBN, with graphene mobility values approaching $\sim 10^4 \text{ cm}^2\text{V}^{-1}\text{s}^{-1}$. We believe that demonstrating CVD-compatible fabrication protocols, as the one introduced for the G-OEM, importantly contributes to the scalability effort toward 2D materials' technology.

In the manuscript, we added the following discussion paragraph regarding scalability:

“Finally, let us comment on the perspective for scalability of the proposed G-OEM. We stress that the CVD graphene employed as channel and gate material is obtained via growth and transfer methods with established wafer-scale capabilities³⁷. The carrier mobility necessary for effective G-OEM operation ($>10^4 \text{ cm}^2\text{V}^{-1}\text{s}^{-1}$ at room temperature), crucially requires decoupling of the graphene channel from extrinsic scattering sources⁹³. As demonstrated both for exfoliated graphene flakes^{55,66}, and CVD-grown graphene single crystals^{94,95}, screening of substrate roughness is paramount. The only reliable strategy currently available to achieve this requires the use of thick hBN⁹⁵. Although large-scale growth of single or few layer single-crystalline hBN is well established, the few-layer thickness limit does not guarantee adequate screening, resulting in carrier mobility in line to that of graphene on SiO_2 ($<10^4 \text{ cm}^2\text{V}^{-1}\text{s}^{-1}$)⁹⁶. The synthesis of scalable counterparts of thick hBN flakes (i.e., providing an adequate screening effect) is a sensitive topic. Relevant advances in crystals dimensions can be found in⁹⁷, where a $2\times 5\text{cm}^2$ trilayer hBN crystals growth has been demonstrated. Similarly, the growth of large area (5 cm lateral dimension) thick (5 nm) hBN have been shown in⁹⁸, with graphene mobility values approaching

$\sim 10^4 \text{ cm}^2\text{V}^{-1}\text{s}^{-1}$. We showed that the presented G-OEM fabrication protocol is compatible with CVD-grown materials (in this case, graphene itself), making it ready for the integration of large-scale thick hBN films.”

Remark n.4

The shown speed and device performance is nothing outstanding for anyone familiar with graphene photodetectors. Nevertheless, the combination of a high-speed graphene device in such a communication link is new to my knowledge and thus worth to publish.

Answer n.4:

We thank the reviewer for recognizing the value of our approach and the novelty arising from our new proposed scheme. Concerning the high speed, we agree with the reviewer that graphene photodetectors can perform much better in terms of operating frequency. Indeed, our setup does not show any roll-off at 96 GHz, this last being the maximum frequency observable in our setup. We stress that the choice of the carrier frequency adopted in this work has been dictated by relevant application perspectives suggested by our industrial partner (Ericsson), even though this value is far from the physical limit of graphene. The possibility to operate graphene detectors up to 500 GHz, as recently demonstrated in *Ref. 46 (new version of the manuscript)* confirms the value of this work, since this achievement suggests that optoelectronic mixers with operating frequencies up to 500 GHz can be realized. Due to this recent advancement, we updated the phrase:

“Frequencies theoretically approaching 500 GHz”.

With:

“frequencies of 500 GHz⁴⁶”.

and the phrase:

Experimental measurements revealed a response time of 2.1 ps, corresponding to a bandwidth of $\sim 262 \text{ GHz}^{45}$, while theory predicts a 3-dB bandwidth exceeding 500 GHz⁴⁴.

With:

Recent direct bandwidth measurements have shown $>500 \text{ GHz}$ bandwidth⁴⁶, according to theory predicting a 3-dB bandwidth exceeding 500 GHz⁴⁴

Remark n.5

Small error in figure caption: fig. 4. C) “active area cross section”

Answer n.5:

We thank the reviewer for having noticed the typo, we modified the figure accordingly.

Reviewer #2

In this paper, the authors report a sub-THz wireless transmission based on graphene integrated optoelectronic mixer, where a bandwidth >96GHz and -44 dB upconversion efficiency are achieved, paving the way to the development of novel arrayed-antennas for millimeter-wave technology relying on a new approach taking advantage of optics.

Therefore, the paper will be of interest to the community and I recommend this article to be published in Nature Communications after answering the following questions:

Remark n.1

The bottom graphene is encapsulated by hBN. The functionality of the hBN need to be clarified. Moreover, since the hBN is mechanically exfoliated from bulk crystals and cannot be easily repeated in a mount of samples, why not use a CVD grown dielectric material?

Answer n.1:

We thank the Reviewer for the comment. It is well established since a few years that hBN employed as graphene substrate substantially reduces carrier scattering [Ref. 65 (new version of the manuscript)] compared to any other dielectric material. Optimal results are obtained when full encapsulation in dry conditions is performed [Ref. 66 (new version of the manuscript)]. Crucially, relatively thick hBN flakes (>15 nm according to Ref. 35 (old and new version of the manuscript)) need to be employed to screen the roughness of the underlying substrate and guarantee sufficiently high carrier mobility.

In the manuscript, we now say:

Relatively thick hBN (>15 nm according to³⁵) is known to crucially screen the roughness of the underlying substrate, resulting in enhanced carrier mobility in graphene⁶⁵, with optimal results upon full encapsulation in dry conditions⁶⁶.

CVD graphene single crystals like the ones used in our work are defect-free (see Ref. 51 in the old version of the manuscript, now Ref. 52), meaning that the carrier mobility is limited by extrinsic disorder [Ref. 93 (new version of the manuscript)], induced by the substrate and/or the transfer process. While over the years the latter has been largely optimized – with wafer-scale compatible protocols currently available [Refs. 36-37 (old and new version of the manuscript)] – no substantial progress has been made in establishing dielectrics substrates alternatives to hBN. Although monocrystalline hBN films can be grown by different techniques, their mono-to-few-layer thickness does not guarantee adequate screening of graphene, nor significant mobility improvement (see, e.g., Ref. 66 (new version of the manuscript)). A recent, promising advance [Ref. 98 (new version of the manuscript)] has shown the growth of 5nm thick hBN with 5cm lateral size. Nevertheless, for the time being the only realistic approach is to rely on exfoliated hBN flakes, i.e., work at the proof-of-concept level, and develop fabrication protocols that can be readily transferred to all-CVD systems in the near future.

In the manuscript, we added the following discussion paragraph regarding scalability:

“Finally, let us comment on the perspective for scalability of the proposed G-OEM. We stress that the CVD graphene employed as channel and gate material is obtained via growth and transfer methods with established wafer-scale capabilities³⁷. The carrier mobility necessary for effective G-OEM operation ($>10^4 \text{ cm}^2\text{V}^{-1}\text{s}^{-1}$ at room temperature), crucially requires decoupling of the graphene channel from extrinsic scattering sources⁹³. As demonstrated both for exfoliated graphene flakes^{65,66}, and CVD-grown graphene single crystals^{94,95}, screening of substrate roughness is paramount. The only reliable strategy currently available to achieve this requires the use of thick hBN³⁵. Although large-scale growth of single or few layer single-crystalline hBN is well established, the few-layer thickness limit does not guarantee adequate screening, resulting in carrier mobility in line to that of graphene on SiO₂ ($<10^4 \text{ cm}^2\text{V}^{-1}\text{s}^{-1}$)⁹⁶. The synthesis of scalable counterparts of thick hBN flakes (i.e., providing an adequate screening effect) is a sensitive topic. Relevant advances in crystals dimensions can be found in⁹⁷, where a 2x5cm² trilayer hBN crystals growth has been demonstrated. Similarly, the growth of large area (5 cm lateral dimension) thick (5 nm) hBN have been shown in⁹⁸, with graphene mobility values approaching $\sim 10^4 \text{ cm}^2\text{V}^{-1}\text{s}^{-1}$. We showed that the presented G-OEM fabrication protocol is compatible with CVD-grown materials (in this case, graphene itself), making it ready for the integration of large-scale thick hBN films.”

Remark n.2

The material of the G-OEM waveguide is SiN, and it can be easily fabricated with Si waveguide based on SOI platform. The authors need to explain the benefit of SiN waveguide in both performance and cost.

Answer n.2:

We thank the reviewer for this comment, which allowed us to clarify this point. Indeed, SOI technology could be a valid alternative to realize OEM functionalities based on graphene. Silicon waveguides would allow to further shrink the device dimensions due to higher index contrast providing larger mode confinement. Nevertheless, Silicon waveguides suffer from two-photon-absorption at relatively low power densities. In the proposed perspective (antenna array) a relatively high input power could be required at the input, before being routed and split toward the single element. The SiN technology supports high optical power densities without suffering from two-photon absorption [Ref. 63 (new version of the manuscript)]. Moreover, the routing of the local oscillator toward each antenna could require a relatively long path. SiN waveguides offer ultra-low propagation loss compared to Silicon waveguides [Ref. 63 (new version of the manuscript)]. Another advantage of SiN technology is the lower thermo-optic coefficient at telecom wavelengths, making SiN photonic circuits more stable against temperature [Ref. 64 (new version of the manuscript)].

We added this phrase in the manuscript:

"It is worth noting that Si waveguides can be used as a valid alternative to SiN waveguides to perform OEM using graphene. Nevertheless, silicon waveguides suffer from two-photon-absorption at relatively low power density. In the proposed perspective (optoelectronic antenna array) a relatively high input power could be coupled at the input of the chip, before being routed and split toward the single element. The SiN technology supports high optical power densities without suffering from two-photon absorption⁶³. Moreover, the routing of the local oscillator toward each antenna could require a relatively long path. SiN waveguides offer ultra-low propagation loss compared to Silicon waveguides⁶³. Another advantage of SiN technology is the lower thermo-optic coefficient at telecom wavelengths, making SiN photonic circuits more stable against temperature⁶⁴. "

Besides optical performances, the choice of the SiN platform has been also dictated by electrical performances considerations: standard SOI technology is based on Si substrates with low resistivity, typically lower than 15 Ωcm . This is detrimental for RF performances, since Ohmic losses in such substrates are non-negligible [Refs. 60,61,62 (new version of the manuscript)] due to the high interaction between the RF mode and the lossy Si substrate. This can be mitigated using Si substrates with high resistivity $> 10\text{K}\Omega\text{um}$ [Ref. 61 (new version of the manuscript)]. An alternative solution is the use of a sufficiently thick oxide to avoid interaction between the electromagnetic RF propagating mode with the substrate. Our SiN technology is based on 15 μm thick BOX (Buried Oxide), allowing to decouple the RF mode from the Si substrate. We added a phrase to clarify this:

"...with negligible ohmic losses. Standard SOI technology is based on Si substrates with low resistivity, typically lower than 15 Ωcm . This is detrimental for RF performances, since ohmic losses in such substrates are non-negligible⁶⁰⁻⁶². RF losses can be mitigated using Si substrates with high resistivity $> 10\text{K}\Omega\text{cm}$ ⁶¹. An alternative solution is the use of a sufficiently thick oxide to avoid interaction between the electromagnetic RF propagating mode with the substrate. Our SiN technology is based on a 15 μm thick BOX (Buried Oxide), allowing to decouple the RF mode from the Si substrate, thus minimizing ohmic losses."

Remark n.3

On top of the Fig 6., the sentence is not complete "... Finally, the inset in ...". The authors need to modify it.

Answer n.3:

We thank the reviewer for having noticed the typo. We wrote the phrase two times. We fixed this issue by removing the repetition.

Remark n.4

In the fabrication method section, the authors should state the details of graphene geometry made.

Answer n.4:

We thank the Reviewer for the suggestion. We added the following paragraph to the Methods section:

“The graphene is patterned by electron beam lithography followed by oxygen plasma etching into a $50 \times 8 \mu\text{m}^2$ rectangle. Two contacts are fabricated on top of graphene (with $2 \mu\text{m}$ superposition on each side, resulting in a final channel of $50 \times 4 \mu\text{m}^2$) using electron beam lithography followed by thermal evaporation of Ni/Au and lift-off (step 7).”

Remark n. 5.1

The equivalent circuit diagram of the G-OEM should be clarified. Also, the authors should theoretically discuss and calculate the bandwidth limit of the system.

Answer n. 5.1:

We thank the reviewer for this question, which allows us to better clarify our circuit model. The circuit shown in the supplementary information keeps the intrinsic behavior of our device, which can be modeled by purely resistive elements in the considered bandwidth range (<100GHz).

Intrinsic bandwidth limitations due to the material (graphene) would be observed at frequencies well above our experiment (~500 GHz), as recently experimentally demonstrated [Ref. 46 (new version of the manuscript)]. Since our device operates well below this bandwidth limitation, we haven't included capacitive contributions modelling the characteristic response time of the material.

Another factor that may limit bandwidth is the design of our device. The metal transmission line embedding the device (which is, in our case, a coplanar transmission line in ground-signal-ground configuration) limit can arise from (i) impedance mismatch between the generator and the transmission line, or between the transmission line and the load, or (ii) ohmic loss due to the interaction between the RF mode and the resistive silicon substrate. In our device the transmission line is designed to have a characteristic impedance of 50Ω in the whole operating frequency range. In addition, our SiN layer is deposited on $15 \mu\text{m}$ BOX (buried oxide), that is sufficient to decouple the RF mode from the Si substrate. Thus, ohmic losses are negligible.

We clarified these points in the discussion contained in the Supplementary information, by adding the following sentences:

“The circuit model is purely resistive and describes the device including metal/graphene contact resistance. This model completely keeps the behavior of the device and is sufficient to describe the optoelectronic mixing operation in the considered frequency range (<100 GHz). The graphene intrinsic dynamics limiting the frequency response of our device would be observed at frequencies well above our experiment (~500 GHz) as very recently shown [2] and therefore is not included in the model. Beside this, the passive RF circuitry embedding the device could lead to bandwidth degradation if not well designed. As shown in Sec. B, the transmission line embedding our device is a coplanar waveguide designed to have a characteristic impedance of 50Ω in the whole operating frequency range.”

We then added the following phrase in the main text:

“Our SiN technology is based on a $15 \mu\text{m}$ thick oxide, allowing to decouple the RF mode from the Si substrate, thus minimizing Ohmic losses.”

Remark n. 5.2

An expectation of how to improve the system is also needed since the graphene experimental measurements revealed a response time of 2.1ps.

Answer n. 5.2

We thank the reviewer for this clarification request. Indeed, as pointed out in the text, our device could achieve a frequency operation greater than the 96 GHz limit of our experimental setup. We do not observe any roll-off up to this frequency. The real bandwidth of the device could be detected using dedicated setups providing access to higher frequency, which are not available within the equipment at our disposal. We stress that the choice of the carrier frequency adopted in this work has been dictated by relevant application

perspectives suggested by our industrial partner (Ericsson), even though this value is far from the physical limit of graphene. A very recent paper showed >500 GHz [Ref. 46 (new version of the manuscript)] bandwidth. This Ref. is now included in the manuscript:

Recent direct bandwidth measurements have shown >500 GHz bandwidth⁴⁶, according to theory predicting a 3-dB bandwidth exceeding 500 GHz⁴⁴

We also integrated the text with a more explicit phrase on the limitation introduced by our setup:

“The result is shown in Fig. 6a and reveals a flat frequency response over the entire frequency window accessible using our measurement system. No roll-off was measured up to 96 GHz, meaning that the operating bandwidth of our device is larger than this value.”

Remark n.6

As the G-OEM is promising to be used in arrayed-antennas, the fabrication method in this article is rather complicated. The authors could discuss how to push this G-OEM to mass-production.

Answer n.6:

We thank the Reviewer for the input. The high-end goals of our research require elevated carrier mobility: currently, the coupling of graphene to hBN is the only reliable approach available in the community. We stress that, although the G-OEM presented in this work includes hBN flakes, the graphene acting both as active channel and gate is CVD grown and transferred employing established wafer-scale techniques [Ref. 37 (old and new version of the manuscript)]. As such, we believe that we do take a relevant step toward mass production. In addition, by using CVD graphene, we demonstrate that the van der Waals assembly techniques presented in the manuscript are suitable for CVD materials. We expect to integrate CVD-grown large-scale thick hBN soon in our process.

In the manuscript, we discuss this point in the following discussion paragraph regarding scalability:

“Finally, let us comment on the perspective for scalability of the proposed G-OEM. We stress that the CVD graphene employed as channel and gate material is obtained via growth and transfer methods with established wafer-scale capabilities³⁷. The carrier mobility necessary for effective G-OEM operation ($>10^4 \text{ cm}^2\text{V}^{-1}\text{s}^{-1}$ at room temperature), crucially requires decoupling of the graphene channel from extrinsic scattering sources⁹³. As demonstrated both for exfoliated graphene flakes^{65,66}, and CVD-grown graphene single crystals^{94,95}, screening of substrate roughness is paramount. The only reliable strategy currently available to achieve this requires the use of thick hBN³⁵. Although large-scale growth of single or few layer single-crystalline hBN is well established, the few-layer thickness limit does not guarantee adequate screening, resulting in carrier mobility in line to that of graphene on SiO_2 ($<10^4 \text{ cm}^2\text{V}^{-1}\text{s}^{-1}$)⁹⁶. The synthesis of scalable counterparts of thick hBN flakes (i.e., providing an adequate screening effect) is a sensitive topic. Relevant advances in crystals dimensions can be found in⁹⁷, where a $2 \times 5 \text{ cm}^2$ trilayer hBN crystals growth has been demonstrated. Similarly, the growth of large area (5 cm lateral dimension) thick (5 nm) hBN have been shown in⁹⁸, with graphene mobility values approaching $\sim 10^4 \text{ cm}^2\text{V}^{-1}\text{s}^{-1}$. We showed that the presented G-OEM fabrication protocol is compatible with CVD-grown materials (in this case, graphene itself), making it ready for the integration of large-scale thick hBN films.”

Reviewer #3

A manuscript by A.Montanaro with co-authors discusses a 92-96 GHz wireless data link, where the focus is made on electro-optical mixer made of electrostatically gated graphene.

The manuscript is very interesting and authors have obviously made novel, large and interesting experimental work. The authors use a novel approach of up-conversion of the baseband signal (1-4GHz) into a sub-THz spectral range (here, 92-96GHz) utilized an LO generated in the same graphene channel as a beating of two lasing lines in the 1.55 μ m wavelength range. This way the authors manage to avoid any additional photonic or THz modulators, hence reducing the device complexity significantly. It is very interesting to see an application demonstration in a sub-THz data link in a room scale (2 meters). A presentation of the graphene electro-optical mixer (G-OEM) up-conversion efficiency based on internal graphene parameters was also done (see some of my remarks).

I find this work sufficiently novel, and interesting. With an overall positive view on this manuscript I have some questions, answering which would, in my opinion, improve the quality of the work. I sort my questions based on the category rather than on sections in the manuscript.

Remark n.1

Authors bring to our attention possible improvements of the G-OEM from the current stage. E.g. an up-conversion efficiency increase by 11dB (see e.g. Discussion). However, based on page 8 in Supplementary Materials, the aforementioned 11dB improvement is justified if the contact resistance to the graphene channel is reduced (e.g. down to 500 $\Omega\mu$ m). Well, on page 9 in SM, authors present performance simulation if the carrier mobility reduces from the current 27000 down to a hypothetical 4000, observing a 15dB efficiency reduction. However, we do not know what can we expect from a , possible, mobility increase above 27000.

Answer n.1:

We thank the reviewer for asking for clarifications on this point, since this allows us to better describe our prediction in terms of performance. As pointed out by the reviewer, we calculate the improvement rising from the reduction of the contact resistance, by considering a uniform distribution of the optical power. We find out that a realistic boost in performance of 11 dB can be achieved, with very small design efforts, as stated in the main text. This would allow to reach (state of the art) performances of concurrent technologies based on the discrete coupling of UTC PDs with modulators. Our model can also predict the value of conversion efficiency against mobility. Such simulation is not trivial to implement, as the microscopic physical mechanisms limiting transport and hot electrons thermalization in low mobility graphene are completely different with respect to the mechanisms involved in hBN-encapsulated graphene. Our simulation accounts for these differences, as described in the supplementary information. Specifically, in our model we tune the cooling terms of the heat equation using effective macroscopic quantities keeping the rich microscopic physics of hot electrons in graphene in different regimes (supercollision cooling in low mobility graphene and cooling by hyperbolic phonons in hBN-encapsulated high mobility graphene). By comparing our model with the experimental data at our disposal and with experimental data found in literature, we find a quite good agreement in both low mobility regime and high mobility (up to 27000 cm²/Vs) regime, and the result is a performance improvement from low to high mobility. We could then go further and calculate the conversion efficiency at higher mobility values (>30000cm²/Vs), but the amount of experimental data available in literature for ultra-high mobility samples are insufficient to be compared with our predictions. For this reason, we think that the calculation of the conversion efficiency in this ultra-high mobility regime could be speculative rather than realistic, as pointed out in the supplementary information: *"Therefore, even if a further increase of mobility could lead to higher hot electrons temperatures, this may not correspond to a net enhancement of conductivity change. Future experimental studies on ultra-high mobility graphene bolometers could clarify this."*

For this reason, we prefer to present a realistic prediction based on the optimization of our device by keeping the same mobility. We stress that this realistic improvement alone, already brings G-OEM performances to values that are directly comparable with SOTA.

We clarify in the main text being more explicit:

“Our simulations show that high mobility ($27.000\text{ cm}^2\text{V}^{-1}\text{s}^{-1}$) is beneficial compared to low mobility ($<10.000\text{ cm}^2\text{V}^{-1}\text{s}^{-1}$), in agreement with experimental results present in literature^{41,42}. We then show that a reduction of contact resistance down to $500\ \Omega\mu\text{m}$ can allow a performance boost....”

Remark n.2

Related to the previous question. For a reduction of the contact resistance to the graphene channel (down to $500\ \Omega\mu\text{m}$) authors give a reference [18], authored by basically the same team. My question is the following, why, holding the technology for a lower R_c , why this technology was not used in the current work?

Answer n.2:

We thank the Reviewer for the question. In *Ref. 18 of the old version of the SI (now Ref. 19 of the SI) ([Ref. 37 of the old and new version of the manuscript])*, some of the authors previously showed that R_c can be as low as $\sim 500\ \Omega\mu\text{m}$, using the same thermally evaporated metal bilayer (Ni/Au) on identical graphene crystals (from a different growth batch). However, R_c strongly depends on doping of the graphene sheet. By means of the current metal contact deposition recipe, the value of $\sim 500\ \Omega\mu\text{m}$ is achieved only at a high concentration level ($\sim 200\text{ meV}$) compared to the 120 meV used in the G-OEM discussed in the manuscript. As shown in the Figure below (Figure 5f from Ref. [19] SI), R_c at $\sim 120\text{ meV}$ is in line with the values obtained for the G-OEM in the current work ($R_c = 1.2 - 1.7\ \text{k}\Omega\mu\text{m}$).

Figure Redacted

We modified the text in the SI file accordingly:

“Considering a contact resistance of $\sim 500\ \Omega\mu\text{m}$, achievable at the wafer scale level (although currently only achievable at high carrier concentration level compared to the operating point of our G-OEM) [19], ...”

Remark n.3

A large emphasize is placed on an array of the discussed devices, Fig.3 a discussion around it. On the other hand, both layers of graphene (the channel and the top-gate) as well as hBN were made by exfoliation. I think authors have to present possible scenarios for the device fabrication which are compatible with the definition “wafer-scale”.

Answer n.3:

We thank the Reviewer for the comment. As we are introducing a new building block for wireless transmission, the scalability is relevant to our research. Nonetheless, we must clarify part of the Reviewer’s argument: the graphene used in this work is exclusively CVD grown (as stated several times in the text). The graphene crystals are grown in matrices on Cu foil and transferred by using the scalable methods reported in

Ref. 37 (old and new version of the manuscript), which legitimates the statement “based on wafer-scale high-mobility graphene”.

According to our experience – as well as to numerous recent papers, starting from the seminal works from Hone group [*Ref.94 (new version of the manuscript)*] and Stampfer group [*Ref. 69 (new version of the manuscript)*] – the transport properties of CVD-grown graphene single crystals are essentially limited by their substrate. To achieve high carrier mobility at room temperature ($>10^4 \text{ cm}^2\text{V}^{-1}\text{s}^{-1}$), screening of substrate-induced roughness by relatively thick hBN flakes ($>15 \text{ nm}$ according to *Ref. 35 (new and old version of the manuscript)*) is essential. While monocrystalline hBN films can be grown by different approaches on wafer scale, their mono-to-few-layer thickness does not guarantee adequate screening of graphene, nor significant mobility improvement (see e.g. [*Ref. 66 (new version of the manuscript)*]). Thick ($>15\text{nm}$) hBN remains the only dielectric environment guaranteeing a mobility of $>10^4 \text{ cm}^2\text{V}^{-1}\text{s}^{-1}$. We added in the manuscript the references to relevant, recent advances towards the synthesis of scalable counterparts of thick hBN flakes (i.e., providing an adequate screening effect). These are [*Ref. 97 (new version of the manuscript)*], which shows a $2\times 5\text{cm}^2$ trilayer hBN crystals growth, and [*Ref. 98 (new version of the manuscript)*], showing the growth of large area (5 cm lateral dimension) thick (5 nm) hBN, with graphene mobility values approaching $\sim 10^4 \text{ cm}^2\text{V}^{-1}\text{s}^{-1}$. We believe that demonstrating CVD-compatible fabrication protocols, as the one introduced for G-OEM, importantly contributes to the scalability effort toward 2D materials' technology. The lack of a large-scale counterpart of exfoliated hBN flakes is regarded as the major holdback for high-end technological applications of graphene. Despite recent advancements [*Ref. 98 (new version of the manuscript)*], for the sake of the device demonstration we used exfoliated hBN flakes, i.e., for the proof-of-concept and for the development of the-fabrication protocols. We reasonably expect that the large area hBN will soon be available in the community for the integration of the full stack over large areas.

In the manuscript, we added the following discussion paragraph regarding scalability:

“Finally, let us comment on the perspective for scalability of the proposed G-OEM. We stress that the CVD graphene employed as channel and gate material is obtained via growth and transfer methods with established wafer-scale capabilities³⁷. The carrier mobility necessary for effective G-OEM operation ($>10^4 \text{ cm}^2\text{V}^{-1}\text{s}^{-1}$ at room temperature), crucially requires decoupling of the graphene channel from extrinsic scattering sources⁹³. As demonstrated both for exfoliated graphene flakes^{65,66}, and CVD-grown graphene single crystals^{94,95}, screening of substrate roughness is paramount. The only reliable strategy currently available to achieve this requires the use of thick hBN³⁵. Although large-scale growth of single or few layer single-crystalline hBN is well established, the few-layer thickness limit does not guarantee adequate screening, resulting in carrier mobility in line to that of graphene on SiO_2 ($<10^4 \text{ cm}^2\text{V}^{-1}\text{s}^{-1}$)⁹⁶. The synthesis of scalable counterparts of thick hBN flakes (i.e., providing an adequate screening effect) is a sensitive topic. Relevant advances in crystals dimensions can be found in⁹⁷, where a $2\times 5\text{cm}^2$ trilayer hBN crystals growth has been demonstrated. Similarly, the growth of large area (5 cm lateral dimension) thick (5 nm) hBN have been shown in⁹⁸, with graphene mobility values approaching $\sim 10^4 \text{ cm}^2\text{V}^{-1}\text{s}^{-1}$. We showed that the presented G-OEM fabrication protocol is compatible with CVD-grown materials (in this case, graphene itself), making it ready for the integration of large-scale thick hBN films.”

Remark n. 4.1

Authors present diverse experimental set-ups for studying different aspects of the G-OEM. However, I find it extremely important to show experimentally what effect the LO power (hence the laser power) will have on the up-conversion efficiency. Specifically since simulations are done for a 20mW laser power and the measurements for a -13dBm.

Answer n. 4.1

We must clarify part of the reviewer's argument: as pointed out by the reviewer, the simulations have been performed for an input optical power of 20 mW, which corresponds to 13 dBm. We used this value in the simulation precisely to be consistent with the experiment, performed at 13 dBm, not -13 dBm as stated in the reviewer remark. Probably the confusion arises from the small font in Fig. 7, where the “~” symbol can be confused with the “-” symbol. In the new version of the manuscript, all the plots and figures have been revised and the fonts have been increased to allow a more agile reading, also in a printed version.

Remark n. 4.2

Interestingly, EVM data are given for a range of laser power (Fig.8), but not for the efficiency.

Answer n. 4.2

We agree with the reviewer that this can be a useful information for the reader. We have retrieved the information required by the reviewer among the whole experimental dataset taken at the time of the measurement and acquired in the same experimental conditions. Unfortunately, the resulting useful dataset consists in just four points, containing the conversion efficiency versus the input optical power, specifically between 3 dBm and 13 dBm, LO power = [3dB, 6dB, 9dB, 13dB]. However, we think that this information is sufficient to show the trend of the conversion efficiency vs the LO power. More specifically, a slope of ~ 1.9 dB/dB is extracted, consistent with the theoretical expected value of (2 dB/dB) that is the usual linear optical power detection regime. Indeed, in the linear regime, the photocurrent is proportional to the optical power, and the associated photogenerated electrical power is the square of the photocurrent. Thus, there is a square law between the coupled optical power and the photogenerated electrical power. For 13 dBm optical IN, a saturation behavior can be inferred, consistent with Fig. 8b in the main text.

We put a plot of the conversion efficiency versus the local oscillator input power in a new section of the supplementary information, with the following text:

“As a complement to the device RF characterization presented in the main text, we measured the conversion efficiency as a function of the optical LO power, while keeping a fixed electrical IF input power of 0 dBm. The result is shown in Fig.S.9. The slope of the experimental curve is 1.9 dB/dB, consistent with the theoretical expected value of (2 dB/dB) that is the usual linear optical power detection regime. Indeed, in the linear regime, the photocurrent is proportional to the optical power, and the associated photogenerated electrical power is the square of the photocurrent. Thus, there is a square law between the coupled optical power and the photogenerated electrical power. For 13 dBm OL power, a saturation behavior can be inferred, consistent with Fig. 8b in the main text.”

FIG. S.9: Conversion efficiency vs OL power, for 0 dBm input IF electrical power

We also added the following phrase in the main text:

“A characterization of the conversion efficiency versus the optical LO power is also present In Supplementary Information, Section III.”

Remark n.5

I do not find a discussion for of the W-band horn antenna gains as a way to improve link range particularly well done. It is a common knowledge that antenna-to-antenna signal transfer is a function of the antenna gains. But it is seems that authors forget that utilization of on-axis (lenses) and off-axis (mirrors) optical elements can easily make a sub-THz link very efficient, at least on distances over 100-500m. I do not say that authors are wrong with their discussion, but they can make this discussion more professional.

Answer n.5

We thank the reviewer for its suggestion. We have modified the text accordingly:

“In general, sub-THz antennas can be made more efficient by making use of on axis and off-axis elements along the path of the link⁸³⁻⁸⁵. Moreover, the performances of complex telecom radio links such as massive MIMO can be further optimized by implementing advanced techniques using, e.g., reconfigurable reflecting/transmitting surfaces⁸⁶”

Remark n.6

Discussion over the “foot-print” of this device vs other devices. Indeed, in most of areas of communication the device scale, their integration possibilities, have not only effect on the system compactness/price but also on functionality extension, e.g. by making arrays, etc. However, I am skeptical in taking just the active graphene area in the consideration. Despite of possible ways of integration, G-OEM-based link is still a complex device, including the lasers, filters, amplifiers, antennas, etc. Do authors have good arguments how to package (ok, excluding the lasers) into the 0.1mm patch?

Answer n.6

We thank the reviewer for the comment that allows us to clarify our discussion on integration. In the manuscript we first consider an antenna array system made by electronics. In such a system, the single antenna element is of the order of mm^2 . An antenna element of a conventional electronic antenna array comprises several electronic components. These are:

- (i) passive components, like inductors and capacitors and mm-wave transmission lines
 - (ii) active elements, like amplifiers
 - (iii) frequency converters.
- (we assume that these three items can be also stacked vertically)

Our paper deals with (iii), i.e., the frequency converters. We propose an alternative to replace (iii) by using an optoelectronic approach, and we discuss the advantages arising from this replacement. We do not propose to modify/eliminate items (i) or (ii). Therefore, an alternative antenna array based on photonic-aided up/down-converters would keep of course all the passive/active electronic elements present in the conventional electronic approach. This includes all the elements surrounding the up/down converting active area. Therefore, we discuss the replacement of the active area performing frequency conversion based on electronic components, with the active area required to perform frequency conversion based on optoelectronic components. Conversely, we notice also that the more conventional optoelectronic solution (modulator + detector) would indeed have an unacceptable too large footprint, and indeed no solution for such high-level integration is present in literature. Instead, the active area of our device, being $3 \times 50 \mu\text{m}^2$, would enable the integration within small footprints. These dimensions are comparable to the dimensions of the active area required by electronic up/downconverters [Ref. 14 (new and old version of the manuscript), and Refs. 53-54 (new version of the manuscript)]. Moreover, our solution overcomes the routing of the electrical LO, since this routing can be done optically, with a potential performance increase, as discussed in the text.

We clarified this in the text by modifying a sentence in the manuscript:

*“The photonic upconverter footprint is $\ll 0.1 \text{ mm}^2$, considering the active area, without access pads for on chip probing. This active area is of the same order of the active area occupied by electronic up/down converters^{14,53,54}, and is much smaller than any other integrated photonic transmitter based on the scheme in **Error! Reference source not found.b.**”*

Remark n.7

Line 413, instead of Fig.8 it is probably Fig.6?

Answer n.7

We confirm it is Fig. 8, but since we did not put any letter, we agree with the reviewer that this could be misleading. We substituted "Fig.8" with Fig.8a.

Remark n.8

On many occasions letters in figures are way too small to be seen in a printed version.

Answer n.8

We thank the reviewer for having noticed this issue, we modified the figures accordingly.

REVIEWERS' COMMENTS

Reviewer #1 (Remarks to the Author):

From my the side, all my concerns have been answered sufficiently in the response and the new version of the manuscript.

I recommend the manuscript for publication in nature communications.

Reviewer #2 (Remarks to the Author):

The authors have replied all the concerns from me. I think this manuscript deserves a suggestion of acceptance of publish.

Reviewer #3 (Remarks to the Author):

I would like to thank the authors for addressing my questions and critics properly and with great details. In my view, the manuscript is of sufficient quality to be published. Sincerely.